# Variability of the Surface Energy Balance in Permafrost Underlain Boreal Forest

Simone Maria Stuenzi[1,2], Julia Boike[1,2], William Cable[1], Ulrike Herzschuh[1,3], Stefan Kruse[1,3], Luidmila A. Pestryakova[5], Thomas Schneider von Deimling[1,2], Sebastian Westermann[6,7], Evgeniy S. Zakharov[4,5], and Moritz Langer[1,2]

[1]Alfred Wegener Institute Helmholtz Centre for Polar and Marine Research, Telegrafenberg A45, 14473 Potsdam, Germany
[2]Geography Department, Humboldt-Universität zu Berlin, Unter den Linden 6, 10099, Berlin, Germany
[3]Institute of Earth and Environmental Science, University of Potsdam, 14476 Potsdam-Golm, Germany
[4]Institute for Biological Problems of Cryolithozone Siberian Branch of RAS, Yakutsk, Russian Federation
[5]Institute of Natural Sciences, North-Eastern Federal University of Yakutsk, Belinskogo str. 58, 677000 Yakutsk, Russia
[6]Department of Geosciences, University of Oslo, Sem Sælands vei 1, 0316 Oslo, Norway
[7]Center for Biogeochemistry in the Anthropocene, University of Oslo, Sem Sælands vei 1, 0316 Oslo, Norway

**Correspondence:** Simone Maria Stuenzi (simone.stuenzi@awi.de)

**Abstract.** Boreal forests in permafrost regions make up around one-third of the global forest cover and are an essential component of regional and global climate patterns. Further, climatic change can trigger extensive ecosystem shifts such as the partial disappearance of near surface permafrost or changes to the vegetation structure and composition. Therefore, our aim is to understand how the interactions between the vegetation, permafrost, and the atmosphere stabilize the forests and the underlying permafrost. Existing model set-ups are often static or are not able to capture important processes such as the vertical structure or the leaf physiological properties. There is a need for a physically based model with a robust radiative transfer scheme through the canopy. A one-dimensional land surface model (CryoGrid) is adapted for the application in vegetated areas by coupling a multilayer canopy model (CLM-ml v0) and is used to reproduce the energy transfer and thermal regime at a study site (N 63.18946°, E 118.19596°) in mixed boreal forest in Eastern Siberia. An extensive comparison between measured and modeled energy balance variables reveals a satisfactory model performance justifying its application to investigate the thermal regime, surface energy balance and the vertical exchange of radiation, heat, and water in this complex ecosystem. We find that the forests exert a strong control on the thermal state of permafrost through changing the radiation balance and snow cover phenology. The forest cover alters the surface energy balance by inhibiting over 90% of the solar radiation and suppressing turbulent heat fluxes. Additionally, our simulations reveal a surplus in longwave radiation trapped below the canopy, similar to a greenhouse, which leads to a comparable magnitude in storage heat flux to that simulated at the grassland site. Further, the end of season snow cover is three times greater at the forest site and the onset of the snow melting processes are delayed.

# 1 Introduction

Around 80% of the world's boreal forest occurs in the circumpolar permafrost zone (Helbig et al., 2016). Despite little human
interference, and due to extreme climate conditions such as winter temperatures below $-50\,^\circ\mathrm{C}$ and very low precipitation,
the biome is highly sensitive to climatic changes (ACIA, 2005; AMAP, 2011; IPCC, 2014) and thus prone to vegetation
shifts. Boreal forest regions are expected to warm by 4 to $11\,^\circ\mathrm{C}$ by 2100, coupled with a more modest precipitation increase
(IPCC, 2014; Scheffer et al., 2012). Moreover, during 2007-2016 continuous zone permafrost temperatures have increased
by $0.39\,(\pm 0.15)\,^\circ\mathrm{C}$ (Biskaborn et al., 2019; IPCC, 2019). The northeastern part of the Eurasian continent is dominated by
such vast boreal forest - the taiga. Due to its sheer size, the biome is not only sensitive to climatic changes, but also exerts
a strong control on numerous climate feedback mechanisms through the altering of land-surface reflectivity, the emission of
biogenic volatile organic compounds and greenhouse gases, and the transfer of water to the atmosphere (Bonan et al., 2018;
Zhang et al., 2011). The forests are usually considered to efficiently insulate the underlying permafrost (Chang et al., 2015).
The canopy exerts shading by reflecting and absorbing most of the downward solar radiation, changes the surface albedo and
decreases the soil moisture by intercepting precipitation and increasing evapotranspiration (Vitt et al., 2000). Additionally, the
forest promotes the accumulation of an organic surface layer which further insulates the soil from the atmosphere (Bonan and
Shugart, 1989). Changing climatic conditions can promote an increasing active layer depth or trigger the partial disappearance
of the near surface permafrost. Further, extensive ecosystem shifts such as a change in composition, density or the distribution
of vegetation (Holtmeier and Broll, 2005; Pearson et al., 2013; Gauthier et al., 2015; Kruse et al., 2016; Ju and Masek, 2016) and
resulting changes to the below- and within-canopy radiation fluxes (Chasmer et al., 2011) have already been reported. Changes
to the vegetation - permafrost dynamics can have a potentially high impact on the numerous feedback mechanisms between
the two ecosystem components. Increased soil carbon release from thawing permafrost through the delivery of soil organic
matter to the active carbon cycle (Schneider Von Deimling et al., 2012; Romanovsky et al., 2017) is modified by vegetation
changes, which can compensate for carbon losses due to an increased $CO_2$ uptake (as observed at ice-rich permafrost sites in
northwestern Canada and Alaska, (Estop-Aragonés et al., 2018)) or even further accelerate total carbon loss.

These vegetation - permafrost dynamics in Eastern Siberia have been documented through exploratory and descriptive field
studies showing a clear insulation effect of forests on soil temperatures (Chang et al., 2015). Further, the biogeophysical
processes controlling the evolution of the ecosystem have been described by conceptual models (Beer et al., 2007; Zhang et al.,
2011; Sato et al., 2016). Modeling schemes such as Orchidee-Can (Chen et al., 2016), JULES (Chadburn et al., 2015), Lund-
Potsdam-Jena (LPJ DGVM) (Beer et al., 2007), NEST (Zhang et al., 2003) or SiBCliM (Tchebakova et al., 2009), have added
a vegetation or canopy module, with defined exchange coefficients for the fluxes of mass and energy, to their soil modules.
This is feasible to varying levels of complexity and the models are capable of addressing a variety of different aspects such as
forest establishment and mortality (Sato et al., 2016), unfrozen vs. frozen ground and fire disturbances (Zhang et al., 2011) or
the evolution of the vegetation carbon density under diverse warming scenarios (Beer et al., 2007).

While all of these studies have significantly improved our understanding of essential mechanisms in boreal permafrost
ecosystems, it is important to further understand how a dynamic evolution of the forest structure and canopy affects the thermal

state and the snow regime of the ground, especially amid ongoing shifts in forest composition (Loranty et al., 2018). The existing model set-ups are often static or not able to capture important processes such as the vertical canopy structure or the leaf physiological properties which strongly control the energy transfer between the top of the canopy atmosphere and the ground. To our knowledge, so far, none of the existing models is able to capture the important processes of the vertical canopy structure in combination with a physically-based, highly advanced permafrost model. The novel, physically-based model introduces a robust radiative transfer scheme through the canopy for a detailed analysis of the vegetation's impact on the hydro-thermal regime of the permafrost ground below. This allows us to quantify the surface energy balance dynamics below a complex forest canopy and its direct impact on the hydro-thermal regime of the permafrost ground below.

With a tailored version of a one-dimensional land surface model (CryoGrid, Westermann et al. (2016)) we perform and analyze numerical simulations and reproduce the energy transfer and surface energy balance in permafrost underlain boreal forest of Eastern Siberia. CryoGrid has, so far, not included a vegetation scheme but has been used to successfully describe atmosphere-ground energy transfer and the ground thermal regime in barren and grass-covered areas (Langer et al., 2016; Westermann et al., 2016; Nitzbon et al., 2019, 2020). In our study, we have adapted a state-of-the-art multilayer vegetation model (CLM-ml v0, originally developed for the Community Land Model CLM by Bonan et al. (2018)). We tailor and implement this scheme to simulate the turnover of heat, water, and snow between atmosphere, forest canopy and ground. We take advantage of a detailed in-situ data record from our primary study site as well as from a secondary, external study site. These data are used to provide model parameters, as well as for model validation by comparing field measurements with simulation results. The main objectives of this study are

1. To demonstrate the capabilities of a coupled multilayer forest - permafrost model to simulate vertical exchange of radiation, heat, and water for boreal forests.

2. To investigate the impact of the new canopy module on the surface energy balance of the underlying permafrost at a mixed boreal forest site in Eastern Siberia.

## 2 Methods

### 2.1 Study Area

Our primary study site is located south east of Nyurba (N 63.18946°, E 118.19596°) in a typical boreal forest zone intermixed with some grassland for horse grazing and shallow lakes. The forest is rather dense and mixed, with the dominant taxa evergreen spruce (*Picea obovata*, 92%), deciduous larch (*Larix gmelinii*, 7.3%) and some hardwood birch (*Betula pendula*, <1%). The average tree height is $5.5\,\mathrm{m}$ for spruce and $12\,\mathrm{m}$ for larch, respectively. These boreal forest environments experience 6 to 8 months of freezing temperatures reaching extremes of $-62\,°\mathrm{C}$ in winter and up to $35\,°\mathrm{C}$ between May and September. The low annual average temperatures result in continuous permafrost and therefore poorly drained, podzolized and nutrient-poor soils (Chapin et al., 2011). Annual precipitation showed an increasing trend from 1900 until 1990, mainly due to an increase in wintertime precipitation. Between 1995 and 2002, summertime precipitation has decreased by -16.9 mm in August and -4.2

mm in July (see Table 1 in Hayasaka (2011) for further details). The temperature trend from 1970 to 2010 for the Central

Yakutian region is positive for spring, summer and fall and negative for winter (monthly surface temperature quantified using Climatic Research Unit (CRU) TS4.01 data (Harris et al., 2014; Stuenzi and Schaepman-Strub, 2020)). The treeline of Northern Siberia is dominated by the deciduous needleleaf tree genus *Larix* Mill. up to N $72.08°$. *Larix sibirica* Ledeb. from E $60 - 90°$, *Larix gmelinii* Rupr. between E $90 - 120°$ and *Larix cajanderi* Mayr. from E $120 - 160°$ (see Fig. 1). Larch compete effectively with other tree taxa because of its deciduous leaf habit and dense bark. In more southern margins of Eastern Siberia, such as

our study area, larch is mixed with evergreen conifers (Siberian pine (*Pinus sibirica, Pinus sylvestris*), spruce (*Picea obovata Ledeb.*), and fir (*Abies sibirica*)), and hardwood (*Betula pendula Roth., B. pubescence Ehrh., Populus tremula L.*) (Kharuk et al., 2019). Moreover, the ground vegetation is poor in diversity and dominated by mosses and lichens that form carpets. Larch has shallow roots, and grows on clay permafrost soils with an active layer of around $0.7\,\mathrm{m}$ and a maximum wetness of 20-40 %. Evergreen conifers and hardwood both prefer deeper active layers and a higher soil moisture availability (Ohta et al.,

2001; Furyaev et al., 2001; Rogers et al., 2015). This study comprises a secondary site for complementary model validation which is situated $581\,\mathrm{km}$ east of the primary site (for further description see Appendix C).

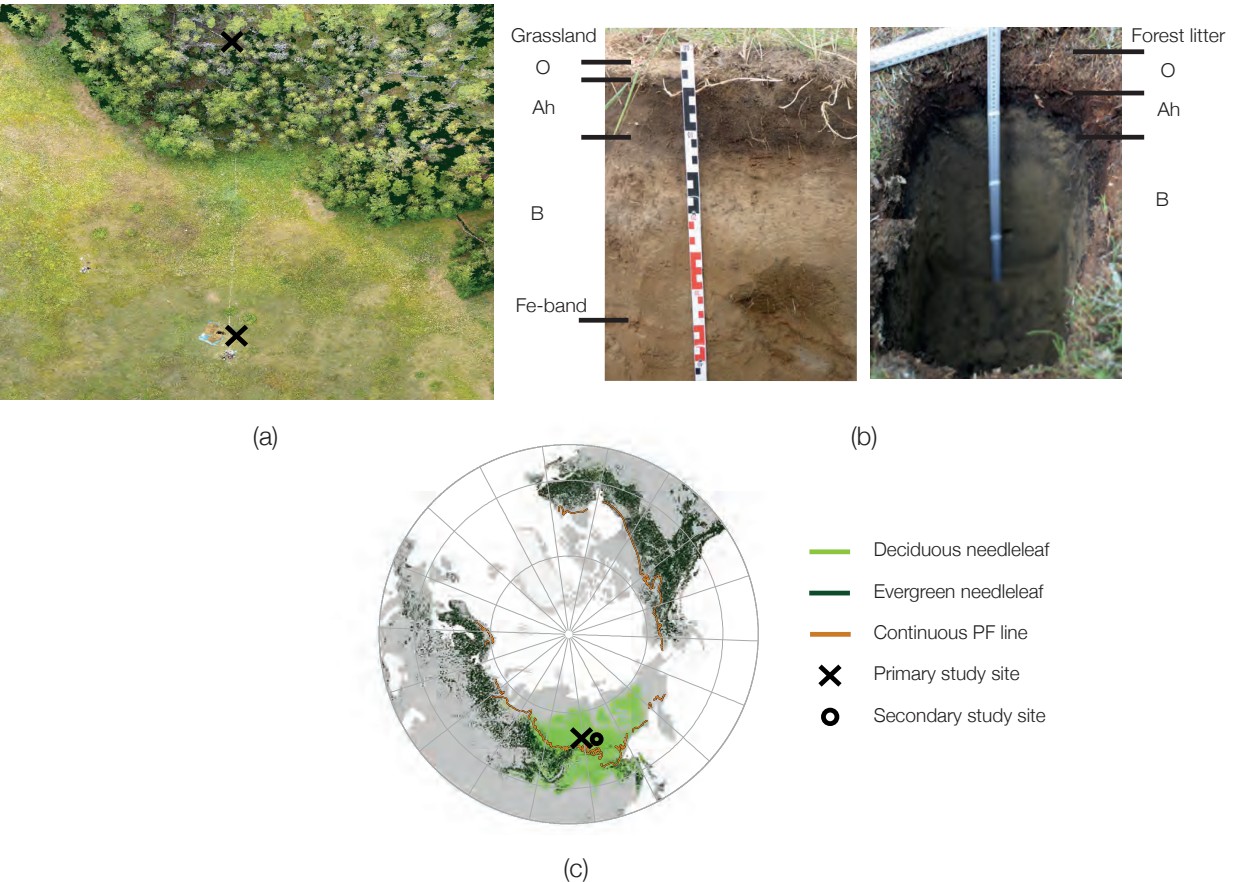

**Figure 1.** (a) Overview image of the location of the automatic weather station (AWS) in the grassland in the lower right corner and the location of the instrumented forest site in the upper left corner (Brieger et al., 2019). (b) Respective soil profiles with the depths of the organic matter dominated O horizon, the top layer of the mineral soil containing the decomposed organic layer (Ah horizon) and the subsoil mineral layer (B horizon) at the grassland site (left) and at the forest site (right). (c) Global evergreen needleleaf (dark green) and summergreen needleleaf (light green) boreal forest distribution and the boundary line between the discontinuous and continuous permafrost extent in brown. The primary study site is marked with a black cross, our secondary study site with a grey circle. Data: ESA CCI Land Cover forest classes. ESA. Land Cover CCI Product User Guide Version 2. Tech. Rep. (2017).

## 2.2 Meteorological and soil physical measurements

An automatic weather station (AWS, Campbell Scientific, detailed list of sensors see Table A1) is installed at $110\,\mathrm{m\,a.s.l.}$ on a meadow next to the forest patch described above. The grassland is grazed by horses in summer and deforestation occurred more then 50 years ago. The AWS records air temperature and relative humidity at two heights ($1.1\,\mathrm{m}$ and $2.5\,\mathrm{m}$) above the ground while wind speed and direction are measured at $3.2\,\mathrm{m}$ above ground. In addition, the station measures liquid precipitation,

snow depth, incoming and outgoing short and longwave radiation and is equipped as a Bowen Ratio station ($B$, see Appendix A). All meteorological variables were recorded with 10 min resolution and stored as 30 min averages. In order to install soil temperature and moisture sensors in the ground a soil pit was excavated in immediate vicinity ($2.5\,\mathrm{m}$) of the AWS. The O horizon has a depth of $0.04\,\mathrm{m}$, the A horizon $0.1\,\mathrm{m}$ containing undecomposed roots, dead moss remains, dense rooting and some organic hummus. The mineral soil is podsolized, sandy and dominated by quartz. The rooting depth is $0.18\,\mathrm{m}$. Iron rich bands were found at $0.4\,\mathrm{m}$, $0.7\,\mathrm{m}$ and $1.1\,\mathrm{m}$. The active layer thickness was $2.3\,\mathrm{m}$ in mid-August 2018 and in early-August 2019. In this soil pit, soil temperature and moisture measurement profiles are installed from the top to the bottom of the active layer consisting of 8 temperature sensors ($0.07\,\mathrm{m}, 0.26\,\mathrm{m}, 0.88\,\mathrm{m}, 1.33\,\mathrm{m}, 1.28\,\mathrm{m}, 1.58\,\mathrm{m}, 1.98\,\mathrm{m}, 2.28\,\mathrm{m}$) and 4 moisture probes ($0.07\,\mathrm{m}, 0.26\,\mathrm{m}, 0.88\,\mathrm{m}, 1.33\,\mathrm{m}$). In addition the conductive ground heat flux in the topsoil layer is measured with a heat flux plate installed at $0.02\,\mathrm{m}$ depth. Further, we record the near surface ground temperature with 5 standalone temperature loggers (iButtons, see Tab. A1) with a measurement interval of 3 hours. These are installed in the upper $0.03\,\mathrm{m}$ of the organic soil at our forest site. The forest soil has a litter layer of $0.08\,\mathrm{m}$ and an organic rich A-horizon reaching a depth of $0.16\,\mathrm{m}$. It is rich in organic and undecomposed material. Mineral soil is podsolized and the rooting depth is $0.20\,\mathrm{m}$. The average active layer thickness between spatially distributed point measurements was $0.75\,\mathrm{m}$ in mid-August 2018 and $0.73\,\mathrm{m}$ in early-August 2019. In a vegetation survey along a $150\,\mathrm{m}$ transect from the grassland into the forest, the tree height of every tree within a $2\,\mathrm{m}$ distance was estimated. Trees <$2\,\mathrm{m}$ were measured with a measuring tape, trees >$2\,\mathrm{m}$ were measured with a clinometer or visually estimated after repeated comparisons with clinometer measurements. Together, the instrumentation with a variety of different loggers, records the spatial and temporal variances across the two sites which are representative for a large area of the mixed boreal forest domain in Eastern Siberia (dataset in review: https://doi.org/10.1594/PANGAEA.914327). In addition, we use a secondary study site, located $581\,\mathrm{km}$ east of Nyurba, with an extensive measurement record of radiation and temperature data from below the forest canopy which allows for a comprehensive model validation (see Appendix C).

## 2.3  Model description

### 2.3.1  Ground module

CryoGrid is a one-dimensional, numerical land surface model developed to simulate landscape processes related to permafrost such as surface subsidence, thermokarst, and ice wedge degradation. The model version is originally described in Westermann et al. (2016) and has since been extended with different functionalities such as lake heat transfer (Langer et al., 2016), multi-tiling (Nitzbon et al., 2019, 2020), and an extensive snow scheme based on CROCUS (Vionnet et al., 2012; Zweigel et al., 2020). The thermo-hydrological regime of the ground is simulated by numerically solving the one-dimensional heat equation with ground water phase change while ground water flow is simulated with an explicit bucket scheme (Nitzbon et al., 2019). The exchange of sensible and latent heat, radiation, evaporation, and condensation at the ground surface are simulated with an surface energy balance scheme based on atmospheric stability functions. In addition, the model encompasses different options to simulate the evolution of the snow cover including the Crocus snowpack scheme. The model is forced by standard meteorological variables which may be obtained from AWSs, reanalysis products, or climate models. The required forcing

variables include air temperature, wind speed, humidity, incoming short-and longwave radiation, air pressure and precipitation (snow- and rainfall) (Westermann et al., 2016). The change of internal energy of the subsurface domain over time is controlled by fluxes across the upper and lower boundaries written as

$$\frac{\delta E}{\delta t} = S_{\text{in}} - S_{\text{out}} + L_{\text{in}} - L_{\text{out}} - Q_{\text{h}} - Q_{\text{e}} - Q_{\text{h}_{\text{precip}}}, \tag{1}$$

where the input to the uppermost grid cell is derived from the fluxes of shortwave radiation ($S_{in}$, $S_{out}$) and longwave ($L_{in}$, $L_{out}$) radiation, at the same time regarding the latent ($Q_e$), sensible ($Q_h$), sensible heat added by precipitation ($Q_{precip}$) and storage heat flux ($Q_s$) between the atmosphere and the ground surface (Westermann et al., 2016).

For this study, we have adapted the multilayer canopy model developed by Bonan et al. (2014, 2018). The canopy model is coupled to CryoGrid by replacing its standard surface energy balance scheme while soil state variables are passed back to the forest module. In the following, we describe the canopy module and its interaction with the existing CryoGrid soil and snow scheme. All differences towards former CryoGrid parameterizations are summarized in Table 1.

**Table 1.** Overview of the processes for which this study differs from the former CG parameterizations.

| Process / Parameter | CG | CG crocus + CLM-ml v0 |
|---|---|---|
| Surface energy balance | *See Eq. 1* | *Surface energy balance modulated by canopy, see Eq. 2, after Bonan et al. (2018)* |
| Precipitation interception | *Direct precipitation from forcing data Westermann et al. (2016)* | *Precipitation modulated by canopy (canopy drip, canopy and stem evaporation, stem flow and direct throughfall), see Eq. 4, after Bonan (2019)* |
| Dynamic evapotranspiration | - | *See Eq. 7 in Bonan et al. (2018)* |
| Snow scheme | *Westermann et al. (2016)* | *Crocus snow scheme. See sect. 2.3.3 after Zweigel et al. (2020)* |

### 2.3.2 Canopy module

The multilayer canopy model provides a comprehensive parameterization of fluxes from the ground, through the canopy up to the roughness sublayer. The implementation of a roughness sublayer allows the representation of different forest canopy structures and their impact on the vertical heat and moisture transfer. The concept is similar to the multilayer approach in ORCHIDEE-CAN (Chen et al., 2016; Ryder et al., 2016). In an iterative manner, photosynthesis, leaf water potential, stomatal conductance, leaf temperature and leaf fluxes are calculated. This improves model performance in terms of capturing the stomatal conductance and canopy physiology, nighttime friction velocity and the diurnal radiative temperature cycle and sensible heat flux (Bonan et al., 2014, 2018). The within-canopy wind profile is calculated using above- and within-canopy coupling with a roughness sublayer (RSL) parameterization (see Bonan et al. (2018) for further detail).

The multilayer canopy model (Bonan et al., 2018) was developed based on the use with CLM soil properties. Following the notations summarized in Bonan (2019) we developed a CLM-independent multilayer canopy module which can be coupled to CryoGrid by integrating novel interactions and an adapted snow cover parameterization. In order to set necessary parameters of the canopy module we make use of values defined for the plant functional type evergreen needleleaf forest of CLM. Please note that all parameters defining the canopy are set as constant values so that vegetation is not dynamic and changes in forest composition or actual tree growth are not considered in this study.

### 2.3.3 Snow module

The snow module employed in this study is based on Zweigel et al. (2020) (submitted, available upon request). The CryoGrid model is extended with a snow microphysics parameterizations based on the CROCUS snow scheme (Vionnet et al., 2012), as well as lateral snow redistribution. The CLM-ml (v0) multilayer canopy model has not yet been coupled to a snow scheme (Bonan et al., 2018). Following Vionnet et al. (2012) the microstructure of the snow-pack is characterized by grain size (gs, mm), sphericity (s, unitless, range 0-1), and dendricity (d, unitless, range 0-1). Fresh snow is added on top of the existing snow layers with temperature and windspeed dependent density and properties. After deposition the development of each layers microstructure occurs based on temperature gradients and liquid water content (Vionnet et al., 2012). Snow albedo for the surface layer and an absorption coefficient for each layer are calculated based on the snow properties. Solar radiation is gradually absorbed throughout the snow layers and the remaining radiation is added to the lowest cell. Additionally, the two mechanical processes of mechanical settling due to overload pressure and wind compression increase snow density and compaction. During snowfall, new snow is added to the top layer in each timestep and mixed with the old snow based on the amount of ice. Once a cell exceeds the snow water equivalent of $0.01\,\mathrm{m}$, which equals a snow layer thickness of $0.03\,\mathrm{m}$, a new snow layer is built. Here, snow accumulates on the ground under the forest canopy. During the first snowfall, the surface energy balance of the ground and snow is calculated for each respective cover fraction. After reaching a snow layer thickness of $0.03\,\mathrm{m}$, the ground surface energy balance is calculated for the snow-pack itself (see Table A2). Variables exchanged based on the snow cover are ground surface temperature, surface thermal conductivity and layer thickness of the layer directly under vegetation. Evaporation flux is subtracted from the snow surface. Top of the canopy wind speed is used to calculate the density of the falling snow. Additionally, snow interception is handled like liquid precipitation interception described in Equation 6.

### 2.3.4 Interactions between the modules

The vegetation module forms the upper boundary layer of the coupled vegetation-permafrost model and replaces the surface energy balance equation used for common CryoGrid representations. The top of the canopy (TOC) surface energy balance is calculated by the vegetation module based on atmospheric forcing. The forest module numerically solves the energy balance of the ground surface below the canopy defined as

$$S_{\mathrm{in_{canopy}}} - S_{\mathrm{out_{ground}}} + L_{\mathrm{in_{canopy}}} - L_{\mathrm{out_{ground}}} - Q_{\mathrm{h_{ground}}} - Q_{\mathrm{e_{ground}}} - Q_{\mathrm{s_{ground}}} = 0, \tag{2}$$

where $L_{\mathrm{in_{canopy}}}$ and $S_{\mathrm{in_{canopy}}}$ are the incoming long- and shortwave radiation at the ground surface and the lower boundary fluxes of the multilayer canopy module, and $S_{\mathrm{out_{ground}}}$ is the outgoing shortwave flux from the ground, $L_{\mathrm{out_{ground}}}$ the outgoing longwave flux, $Q_{\mathrm{h_{ground}}}$ the sensible heat flux, $Q_{\mathrm{e_{ground}}}$ the latent heat flux, and $Q_{\mathrm{s_{ground}}}$ the storage heat flux at the ground surface. The first six components of the sub-canopy energy balance directly replace the respective components surface energy balance scheme of CryoGrid (see Equation 2). Whereas $Q_{\mathrm{s_{ground}}}$ is calculated based on temperatures of the uppermost ground or snow layers that are passed from CryoGrid to the forest module. The storage heat flux is calculated as

$$Q_{\mathrm{s_{ground}}} = k\frac{T_{\mathrm{s}} - T_{\mathrm{ground}}}{\Delta z}, \tag{3}$$

where k is the soil thermal conductivity, $T_{\mathrm{s}}$ the soil surface temperature, $T_{\mathrm{ground}}$ the actual ground temperature for the first layer below the surface and $\Delta z$ the layer thickness. Soil thermal conductivity is parameterized following Westermann et al. (2013, 2016) and is based on the parameterization in Cosenza et al. (2003). The thermal conductivity of the soil is calculated as weighted power mean from the conductivities and volumetric fractions of the soil constituents water, ice, air, mineral and organic (Cosenza et al., 2003). In Fig. 2 the energy fluxes expected for a forested and a grassland site in the snow-covered and snow-free periods each are illustrated schematically.

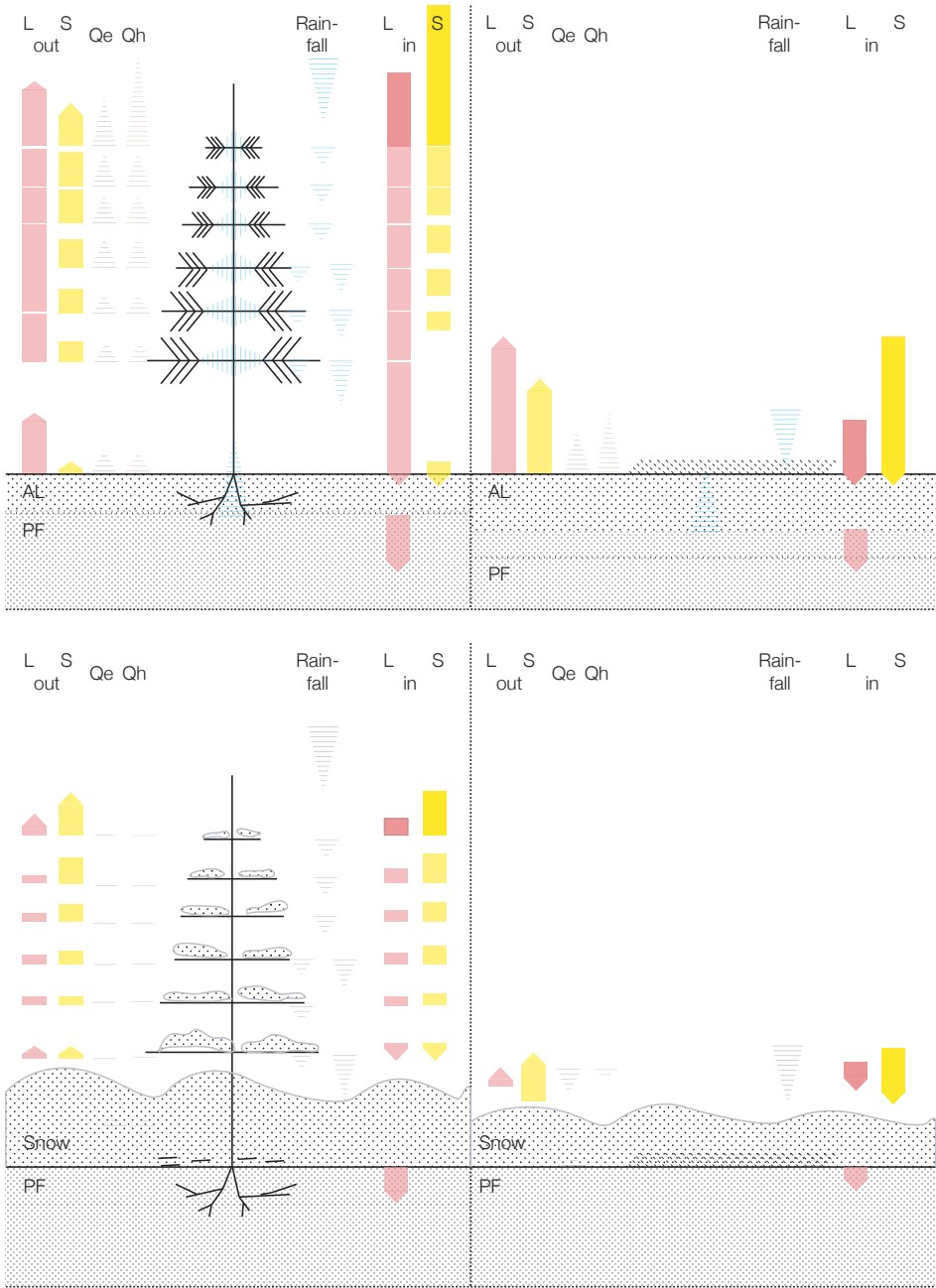

**Figure 2.** Schematic of the surface energy and water balance of the forest (left) and grassland (right) schemes for top, the snow-free period and bottom, the snow-covered period. The active layer (AL), the permafrost (PF), tree and grassland (dotted lines) and the snow-pack (Snow) in the snow covered period. In each of the four panels, incoming and outgoing longwave ($L_{in}$, $L_{out}$), incoming and outgoing solar ($S_{in}$, $S_{out}$), turbulent fluxes ($Q_h$ and $Q_e$), the storage heat flux ($Q_s$), and precipitation (and interception) of rain- and snowfall are shown.

In the novel model set-up which allows for soil-vegetation interaction, the vegetation module receives ground state variables of the top $0.7\,\mathrm{m}$ of the soil layers. These state variables are soil layer temperature ($T_{\mathrm{ground}}$) and soil layer moisture ($W_{\mathrm{ground}}$), as well as the diagnostic variables soil layer conductivity ($k_{\mathrm{ground}}$), and ice content ($I_{\mathrm{ground}}$). The vegetation transpiration fluxes are subtracted from the ground soil layers within the rooting depth and evaporation fluxes from the ground surface. Following the notation in Bonan et al. (2018) the rain and snow fraction reaching the ground ($W_{\mathrm{ground_s}}$) is described as follows

$$\frac{\delta W_{ground_s}}{\delta t} = fP_R + D_c - E_c + D_t - E_t \tag{4}$$

and consist of the direct throughfall ($fP_R$), the canopy drip ($D_c$), the canopy evaporation ($E_c$), the stemflow ($D_t$) and the stem evaporation ($E_t$), which are based on the retained canopy water ($W_c$)

$$\frac{\delta W_c}{\delta t} = (1 - f - f_t)P_R - E_c - D_c, \tag{5}$$

where $1 - f - f_t P_R$ is the intercepted precipitation, and the retained trunk water ($W_t$)

$$\frac{\delta W_t}{\delta t} = f_t P_R - E_t - D_t, \tag{6}$$

respectively. Lateral water fluxes are neglected in this baseline, one-dimensional model set-up.

## 2.4 Model setup and simulations

We run model simulations for forested and non-forested scenarios based on in-situ measurements recorded in 2018 and 2019. The subsurface stratigraphies used in CryoGrid is described by the mineral and organic content, natural porosity, field capacity and initial water/ice content. Some of these parameters could be measured at the forest and grassland sites and were used to set the initial soil profiles in the model. Table A3 summarizes all parameter choices for soil stratigraphies and Table A4 summarizes constants used. The subsurface stratigraphy extends to $100\,\mathrm{m}$ below surface where the geothermal heat flux is set to $0.05\,\mathrm{W\,m^{-2}}$ (Langer et al., 2011b). The ground is divided into separate layers in the model. The uppermost $8\,\mathrm{m}$ have a layer thickness of $0.05\,\mathrm{m}$, followed by $0.1\,\mathrm{m}$ for the next $20\,\mathrm{m}$, $0.5\,\mathrm{m}$ up to $50\,\mathrm{m}$ and $1\,\mathrm{m}$ thereafter. The remaining CryoGrid parameters were adopted from previous studies using CryoGrid (see Table A2) (Langer et al., 2011a, b, 2016; Westermann et al., 2016; Nitzbon et al., 2019, 2020). We use ground surface temperature (GST) as the target variable for model validation. GST results from the surface energy balance at the interface between canopy, snow cover, and ground and provides an integrative measure of the different model components. In addition it is the most important variable determining the thermal state of permafrost.

For the canopy stratigraphy, we follow the parameterizations in Bonan et al. (2018) for the plant functional type evergreen needleleaf (see Table A5 and A6). This canopy stratigraphy can be described by two parameters: the leaf area index (LAI) measured at the bottom of the canopy defines the total leaf area. The leaf area density function on the other hand describes the foliage area per unit volume of canopy space which is the vertical distribution of leaf area. Leaf area density is measured

by evaluating the amount of leaf area between two heights in the canopy separated by the distance. This function can be expressed by the beta distribution probability density function which provides a continuous representation of leaf area for the use with multilayer models (see Bonan (2019) for further information). Here, we use the beta distribution parameters for needleleaf trees (p = 3.5, q = 2) which resembles a cone-like tree shape. LAI can be estimated from satellite data, calculated from below-canopy light measurements or by harvesting leaves and relating their mass to the the canopy diameter. Ohta et al. (2001) have described the monitored deciduous-needleleaf forest site at Spasskaya-Pad research station (our secondary study site, see Appendix C), which has comparable climate conditions but is larch-dominated. The value of the tree plant area index (PAI), obtained from fish-eye imagery and confirmed by litter fall observations, varied between $3.71\,\mathrm{m^2\,m^{-2}}$ in the foliated season and $1.71\,\mathrm{m^2\,m^{-2}}$ in the leafless season. This value does not include the ground vegetation cover. Further, Chen et al. (2005) compared ground-based LAI measurements to MODIS values at an evergreen-dominated study area (57.3° N, 91.6° E) south-west of the region discussed here, around the city of Krasnoyarsk. The mixed forest consists of spruce, fir, pine and some occasional hardwood species (birch and aspen). They find LAI values between 2 and $7\,\mathrm{m^2\,m^{-2}}$. To assess the LAI we use data from literature and the experience from the repeated field work at the described site. Following Kobayashi et al. (2010) who conducted an extensive study using satellite data, the average LAI for our forest type is set to $4\,\mathrm{m^2\,m^{-2}}$ and stem area index (SAI) is set to $0.05\,\mathrm{m^2\,m^{-2}}$, resulting in a plant area index (PAI) of $4.05\,\mathrm{m^2\,m^{-2}}$ and 9 vegetation layers for model simulations. The lower atmospheric boundary layer is simulated by $4\,\mathrm{m}$ of atmospheric layers.

We perform simulations over a 5-year period from August 2014 to August 2019. The model runs are initialized with a typical temperature profile of $0\,\mathrm{m}$ depth: $0°\mathrm{C}$, $2\,\mathrm{m}$: $0°\mathrm{C}$, $10\,\mathrm{m}$: $-9°\mathrm{C}$, $100\,\mathrm{m}$: $5°\mathrm{C}$, $5000\,\mathrm{m}$: $20°\mathrm{C}$. Spin-up period prior to the validation period is 4 years before we compare modeled and measured data. Test runs with a longer spin up period of 10 years confirmed that only 4 years are sufficient when focusing on GST. The meteorological forcing data required by the model include: air temperature, relative humidity, air pressure, wind speed, liquid and solid precipitation, incoming short- and longwave radiation and cloud cover. ERA-Interim data for the coordinate N 63.18946°, E 118.19596° were used to obtain forcing data for the total available period from 1979 to 2019 (Simmons et al., 2007).

## 3 Results

### 3.1 Model validation and in-situ measurements

At our primary study site, the model is validated against ground surface temperature (GST) measurements of forested and non-forested study sites. The data set used covers one complete annual cycle from 10. August 2018 to 10. August 2019. In addition, the model output is compared to radiation, snow depth, conductive heat flux, precipitation and temperature measurements of the AWS at the grassland site. The AWS was set up on 5. August 2018 and taken down on 26. August 2019. Data were recorded continuously, except for 40 days in late May / early June due to a power cut. The mean annual air temperature was $-7.3°\mathrm{C}$ with a maximum temperature of $33.1°\mathrm{C}$ and a minimum of $-54.0°\mathrm{C}$, and an average relative humidity of 70.5%. Precipitation is $129.8\,\mathrm{mm/year}$ (liquid). The maximum snow height at the grassland site is measured to be $0.5\,\mathrm{m}$ in February and the ground was snow-covered for 181 days from 28. October 2018 - 27. April 2019 (values above $0.05\,\mathrm{m}$ snow height). A

quality check of the radiation data revealed partly inconsistent incoming longwave radiation measurements for the time-span of 1. November 2018 - 26. February 2019. During this period it is likely that the sensor is partially covered by snow, making it necessary to discard those measurements from the record. High quality $Q_{net}$ and $L_{in}/L_{out}$ measurements, thus, only exist for the periods 28. to 30. October 2018 and 27. February 2018 to 27. April 2019. This data gap consequently also limits the period for which Bowen Ratios are calculated and sensible heat fluxes ($Q_h$) and latent heat fluxes ($Q_e$) can be derived. The mean annual grassland albedo is $0.35$ with an average of $0.30$ during the snow-free and $0.48$ during the snow-covered season. From December to February the albedo reaches its highest values with a mean of 0.7. Mean annual GST at $0.07\,\mathrm{m}$ depth is $-2.6\,^{\circ}\mathrm{C}$ (range from $19.1\,^{\circ}\mathrm{C}$ to $-24.9\,^{\circ}\mathrm{C}$) with an average of $-11.4\,^{\circ}\mathrm{C}$ in the snow-covered period and $8.0\,^{\circ}\mathrm{C}$ in the snow-free period. The average annual GST recorded in forested areas at a depth of $0.03\,\mathrm{m}$ is $1.9\,^{\circ}\mathrm{C}$ (range from $15.6\,^{\circ}\mathrm{C}$ to $-23.4\,^{\circ}\mathrm{C}$) with an average of $-9.3\,^{\circ}\mathrm{C}$ in the snow-covered period and $5.6\,^{\circ}\mathrm{C}$ in the snow-free period.

We acknowledge that the target variable GST does not allow a detailed evaluation of the surface energy balance. Therefore, we further validate the model performance with additional measurements from an external study site (see Appendix C). Through the Arctic Data Archive system (ADS) we have been provided with meteorological and radiation data from beneath and above the larch-dominated forest canopy in Spasskaya-Pad for 2017-2018 (Maximov et al., 2019). This data is used for additional model validation and is added to the appendix of our manuscript. Overall our analysis reveals a satisfactory agreement between modeled and measured components of the surface energy balance below the canopy. Thus, we argue that the performance of the model at the external study site justifies its application at the primary study site in Nyurba where below canopy fluxes were not acquired (see Appendix C).

### 3.1.1 Surface energy balance

In a first step we assess the surface energy balance by comparing the modeled net radiation ($Q_{net}$), sensible heat flux ($Q_h$), latent heat flux ($Q_e$) and the storage heat flux ($Q_s$) at the forested site and the modeled and measured fluxes at the grassland site (see Fig. 3 and Appendix A). Turbulent fluxes at forest ground are close to zero for both snow-free and snow-covered periods. TOC sensible heat flux is highest in snow-free period ($48.6\,\mathrm{W\,m^{-2}}$), resulting in the highest net radiation flux ($83.5\,\mathrm{W\,m^{-2}}$). Forest ground net radiation flux is only a third ($25.8\,\mathrm{W\,m^{-2}}$) of the TOC flux. Latent heat flux in the snow-free period is similar for forest TOC ($16.5\,\mathrm{W\,m^{-2}}$), and grassland (measured ($22.1\,\mathrm{W\,m^{-2}}$) and modeled ($18.5\,\mathrm{W\,m^{-2}}$)). During the snow-covered season forest TOC and ground turbulent heat fluxes and net radiation are close to zero. Net radiation flux in the snow-covered period is smallest at the grassland site modeled ($-20.0\,\mathrm{W\,m^{-2}}$) and measured ($9.7\,\mathrm{W\,m^{-2}}$). The resulting storage flux is more than double at the forest ground ($30.4\,\mathrm{W\,m^{-2}}$) for the snow-free period and slightly positive ($0.5\,\mathrm{W\,m^{-2}}$) during the snow-covered period.

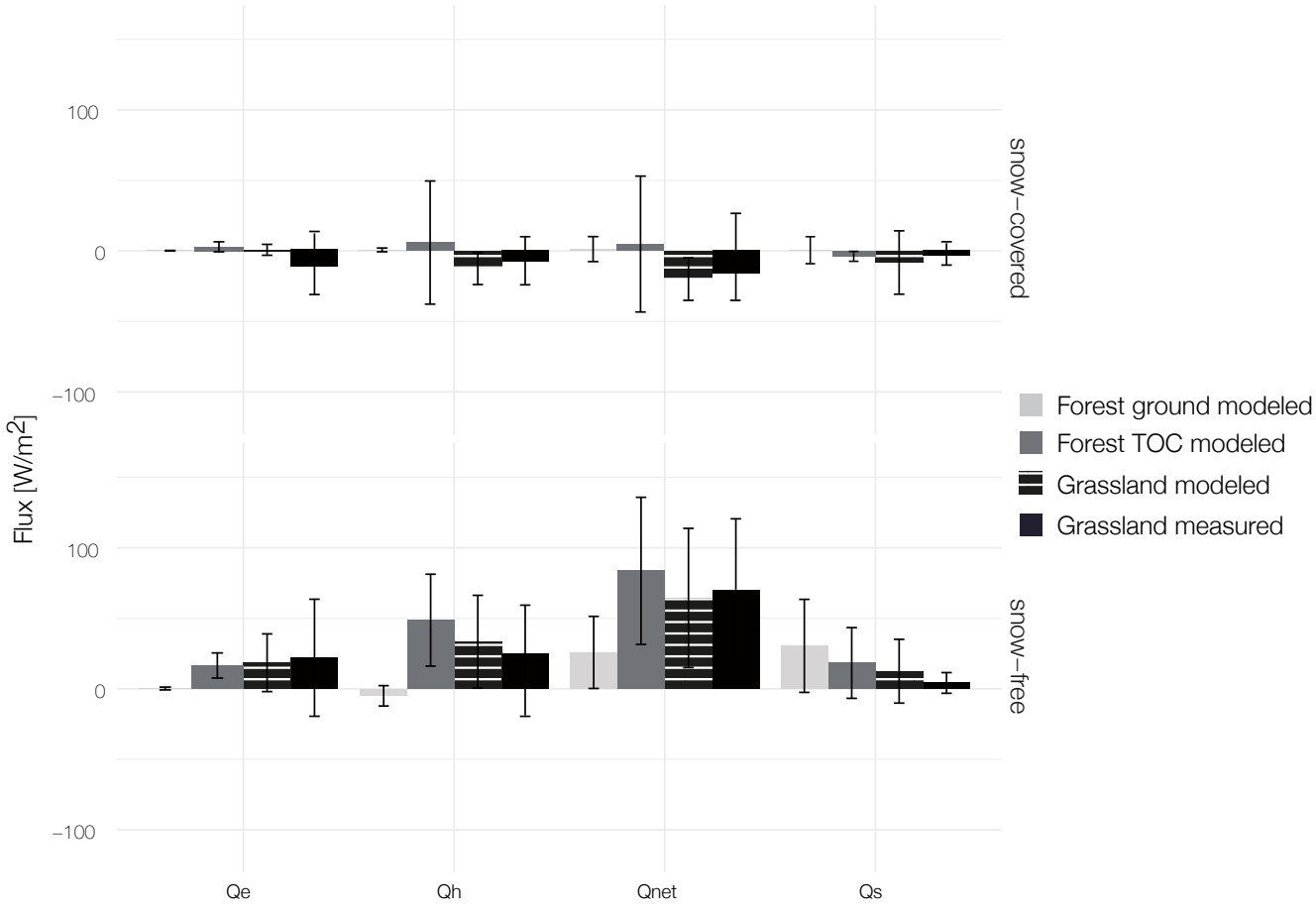

**Figure 3.** Surface energy balance for snow-covered (28.10.2018-27.04.2019) and snow-free (10.10.2019-27.10.2019 and 28.04.2019-10.10.2019) periods at the ground surface of grassland and forest and at the top of the canopy of forest (Forest TOC). Shown are the net radiation ($Q_{net}$), sensible ($Q_h$), latent ($Q_e$) and storage heat flux ($Q_s$) for the model runs of the forest and grassland site as well as the measured values at the grassland site. The bars indicate mean values while the whiskers show the corresponding standard deviations.

Average measured Bowen Ratio ($B$) at the AWS is 1.04, with an average of 1.94 for snow-free and 0.35 for snow-covered
periods. At the grassland site the model predicts a $B$ of 1.8 for the snow-free period and $-16.54$ for the snow-covered period. Which sums up to an annual average $B$ of 1.09 for grassland. Modeled annual average $B$ at the forest ground is more than double with 2.85 for the snow-free period and 2.01 for the snow-covered period. Top of canopy modeled annual average $B$ is 2.99, with an average of $-0.77$ in the snow-covered period and 3.93 in the snow-free period.

More detailed insights into differences of available radiation at the ground surface are presented in Fig. 4. Here, the incoming
short- and longwave radiation measured at the grassland site and modeled for the forest and grassland are shown. The longwave radiation dominates the incoming part of the radiation balance at both sites throughout the year. In the snow-free period downward longwave radiation flux is $44.2\,\mathrm{W\,m^{-2}}$ higher at the forest ground. In the snow-covered period downward longwave

radiation flux is $40.7\,\mathrm{W\,m^{-2}}$ higher at the forest ground. This results in a surplus of energy of $+9.6\,\mathrm{W\,m^{-2}}$ for snow-covered and $+16.7\,\mathrm{W\,m^{-2}}$ for snow-free periods under the forest canopy compared to the grassland. The shortwave radiation reaching

the forest ground is very small for both periods ($9.9\,\mathrm{W\,m^{-2}}$ for snow-free and $2.8\,\mathrm{W\,m^{-2}}$ for snow-covered periods), showing that the canopy effectively intercepts (absorbs and reflects) most of the incoming shortwave radiation. Shortwave down at the grassland site is more than 19 times higher in the snow-free period and 18 times higher in the snow-covered-period.

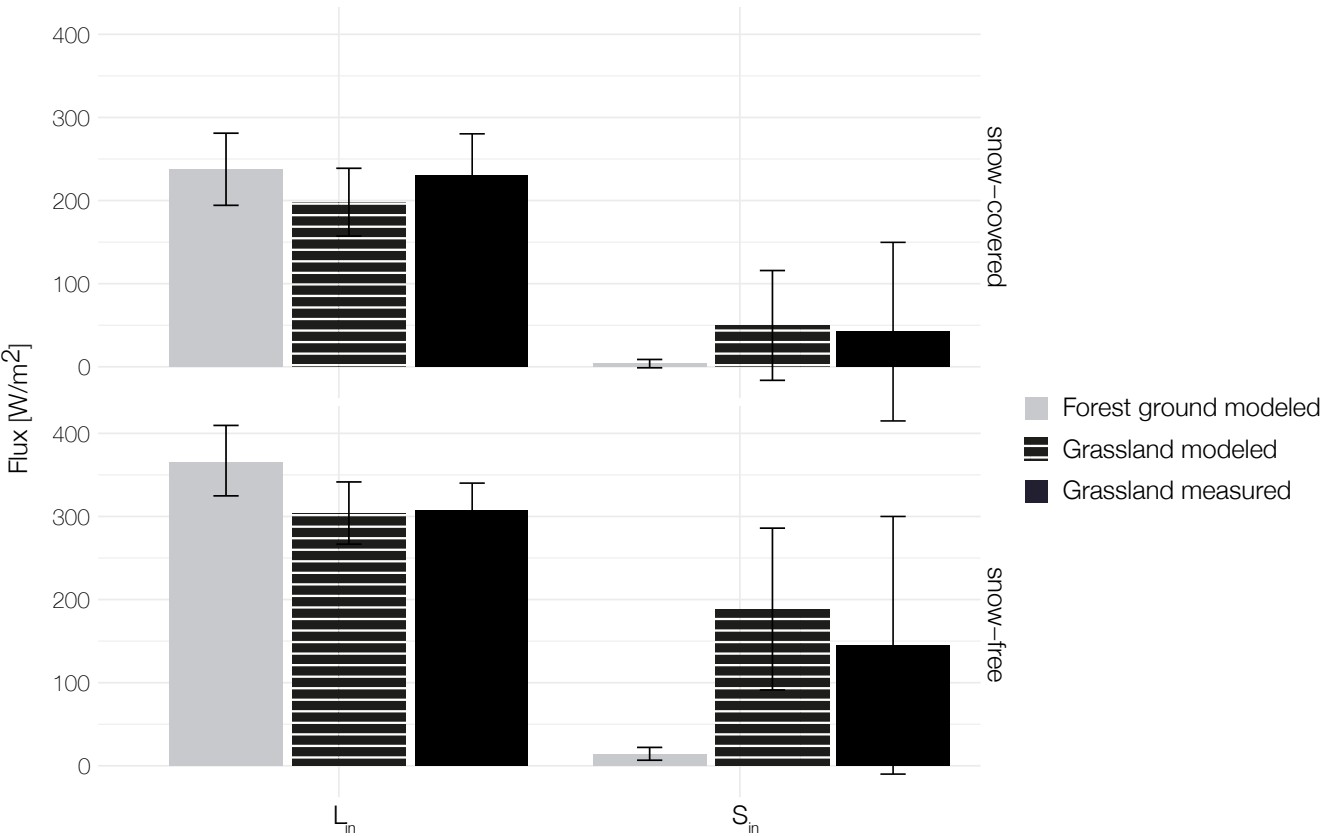

**Figure 4.** Modeled incoming solar and longwave radiation for snow-covered (28.10.2018-27.04.2019) and snow-free (10.10.2019-27.10.2019 and 28.04.2019-10.10.2019) periods at the ground surface of forest (grey) and grassland (striped). Measured (black) incoming solar (for the same time periods) and longwave radiation (for 28. to 30. October 2018 and 27. February 2018 to 27. April 2019) are shown for the grassland site. The bars indicate mean values while the whiskers show the corresponding standard deviations.

### 3.1.2   Thermal regime of the ground near the surface

In a second step, we compare the annual, snow-free and snow-covered period average GST to understand the overall model

performance and the relative temperature differences between the forest and grassland sites (see Fig. 5). We further discuss the

annual cycle of the thermal development of the permafrost ground, the modeled and measured active layer thickness, as well as the volumetric ground water content at both of our study sites.

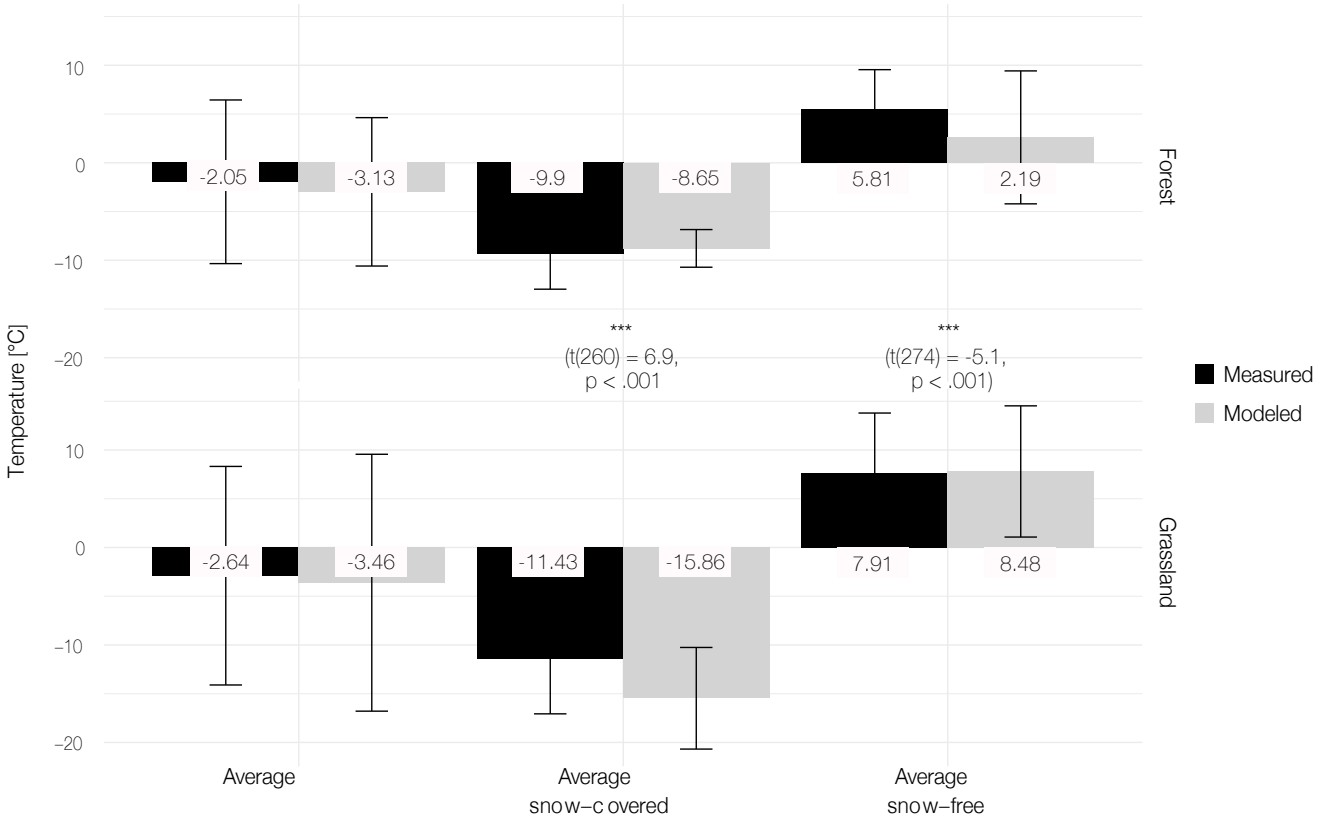

**Figure 5.** Average measured and modeled snow-covered period GST, average measured and modeled snow-free period GST, annual average measured and modeled GST and the respective standard deviations in forest (top, at $0.03\,\mathrm{m}$ depth) and grassland (bottom, at $0.07\,\mathrm{m}$ depth) over a measurement period of 1 year (10. August 2018 - 10. August 2019). The bars indicate mean values while the whiskers show the corresponding standard deviations. Unpaired t-test between modeled forest and grassland GST shows a statistically significant temperature difference for snow-covered and snow-free periods.

Measured and modeled average GST values are summarized in Fig. 5. The highest deviation between modeled and measured GST is found at the grassland site. Here, the model shows a cold bias of $-4.1\,^{\circ}\mathrm{C}$ for the snow-covered period. Also, there is a cold bias of $-2.8\,^{\circ}\mathrm{C}$ in the snow-free period in the forest. Overall, we find an average annual difference of $0.7\,^{\circ}\mathrm{C}$ between the two sites. For snow-free season this difference is $5.2\,^{\circ}\mathrm{C}$ and snow-covered $6.7\,^{\circ}\mathrm{C}$, respectively.

For a more detailed understanding of the annual cycle of the thermal evolution of permafrost ground at our study sites we compare the weekly averaged GST at the grassland and forest site (see Fig. 6). The more detailed analysis of the annual cycle reveals periods with distinct differences between the model simulations and the measured values. For both study sites the model produces a slight GST overestimation in summer and a prolonged thawing period in spring. The measured data shows a much

faster ground warming in spring. This difference is over 20 days at the forest site and 15 days at the grassland site. In addition, there is a cold bias by $5\,°C$ in January at the grassland site. This bias is not seen at the forest site. Thawing starts later in model simulations than measured.

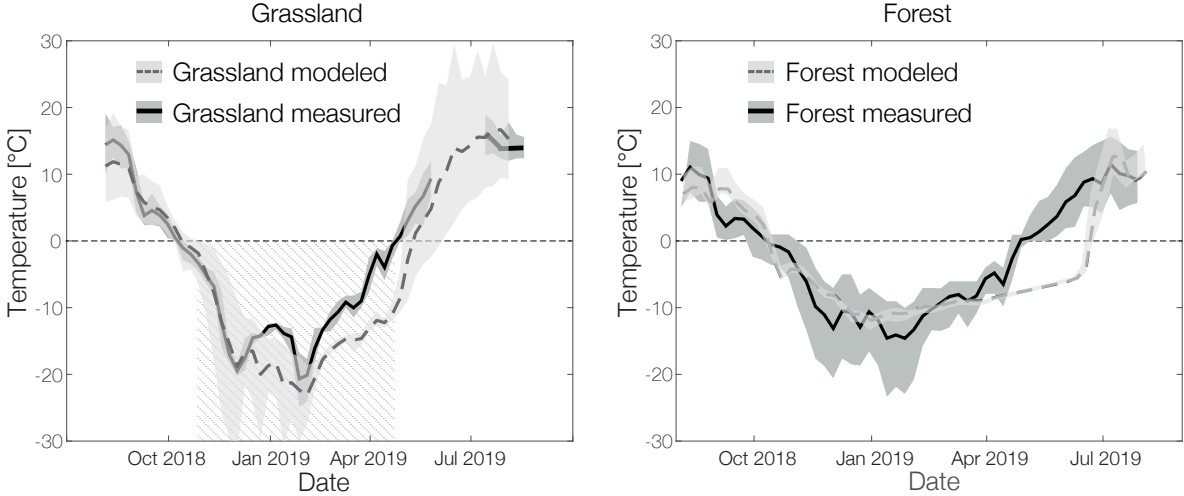

**Figure 6.** Left, modeled (grey) and measured (black) average weekly GST in $0.03\,m$ depth in forest, and right, in grassland in $0.07\,m$ depth with standard deviation for modeled (light grey) and measured (dark grey). In addition, the measured duration of the snow-covered period is shaded at the grassland site. Please note that there is a data gap in the measured data at the grassland site in May-June (see section 3.1 for further details).

To further investigate the temporal evolution of the permafrost ground, we compare modeled and measured active layer
thicknesses at both study sites. In the grassland, the modeled maximum active layer thickness (ALT) is $2.35\,m$ between 13. and 24. October 2018, complete freezing occurred on the 9th of November and top soil thawing started on the 3rd of May. The measured ALT in the grassland was $2.3\,m$ in mid-August 2018 and early-August 2019. The measured ALT in the forest was between $0.5$ and $1.1\,m$ in mid-August 2018. In the forest, the modeled maximum ALT is $2.05\,m$ in October 2018 with freezing being completed on the 14th of November. Top soil thawing begins on June 23rd, 51 days later than in grassland. The modeled
ALT in August 2018 is between $0.4\,m$ and $1.8\,m$ and therefore overestimated by $0.3\,m$ compared to the point measurements taken in August 2018 (see Fig. 7).

Moreover, the measured volumetric water content (VWC) in grassland reaches its maximum of 0.2 in August. The averaged measured VWC at the forest site in August 2018 was 0.3. The model can broadly reproduce this difference but their is a model bias towards higher VWC for both sites. The modeled maximum water content in forest is 0.5 between the end of June and the
330 beginning of July and is about 0.2 higher than in grassland, where the simulation shows a maximum VWC of 0.3 in August. The modeled winter ice content in grassland reaches a maximum value of 0.36 and 0.42 in forest.

## 4 Discussion

The model presented here is found to be capable of simulating the differences in the ground thermal regime between a forested and a non-forested site for permafrost underneath boreal forests. This can provide important insights into the range of spatial differences and possible temporal changes that can be expected following current and future landscape changes such as deforestation through fires, anthropogenic influences and afforestation in currently unforested grasslands or the densification of forested areas. The implemented scheme is able to simulate the physical processes that define the vertical exchange of radiation, heat, water and snow between permafrost and canopy. Our simulations show that the forests exert a strong control on the thermal state of permafrost. At the grassland site, we find a much larger ground surface temperature (GST) amplitude of $60.35\,^{\circ}$C over the annual cycle, which is $32\,^{\circ}$C higher than at the forest site. This vegetation dampening effect on soil temperature is well-described in literature (Oliver et al., 1987; Balisky and Burton, 1993; Chang et al., 2015). Earlier work by Bonan and Shugart (1989) found that forest soils generally thaw later and less deeply, and are cooler than in open areas. In the winter, forested soils are typically warmer relative to open areas. The tree cover can maintain stable permafrost under otherwise unstable thermal conditions (Bonan and Shugart, 1989). Our results are in agreement with these observations, but further demonstrate that the impact of mixed boreal forest on the GST is strongest during the snow period and the summer peak with the warmest months. Our model reveals an average of $6.7\,^{\circ}$C higher GST during the snow-covered period and $5.2\,^{\circ}$C lower GST during the snow-free period. Measurements reveal an average of $2\,^{\circ}$C higher GST in forest during the snow-covered period and $2.3\,^{\circ}$C lower GST during snow-free period. Our model simulations unravel that the strong control on the thermal state of permafrost is a result of the combined effects of canopy shading, suppression of turbulent heat fluxes, below-canopy longwave enhancement, increased soil moisture and distinct snow cover dynamics. These relevant processes controlling forest insulation will be discussed individually in the following subsections followed by a detailed discussion on the model applicability and limitations.

### 4.1 Canopy shading and longwave enhancement

The surface energy balances simulated by the model are very different for grassland and forest. The forest canopy reflects and absorbs over 92% of incoming solar radiation for both snow-free and snow-covered periods. The forest ground albedo therefore has little influence on the energy balance. This canopy shading effect makes the longwave radiation the largest source of radiative energy at the forest site for both time periods. A surplus of longwave radiation by over 20% is largely trapped below the canopy due to extremely low turbulent heat fluxes, similar to a greenhouse. The increased longwave radiation results in a relatively strong storage heat flux. Despite shading, the storage heat flux at the forest site is similar in magnitude to that simulated at the grassland site. This explains the small difference in the modeled depths of the active layer at both sites. The heat flux plate used for measuring the conductive heat flux at the grassland site is designed for use in mineral soil. Due to a certain amount of organic content in the upper soil layer under- or overestimated heat fluxes are possible (Ochsner et al., 2006). This could explain to some extent the difference between measurements and simulations. We further find, that during the snow-free period, the sensible and latent heat flux at the canopy top are high while being close to zero at the forest ground. In summary,

we show that the canopy effectively absorbs and reflects the majority of incoming solar radiation, making canopy shading one of the main controlling mechanisms and that the canopy enhances the longwave radiation the forest ground, because of extremely low turbulent heat fluxes.

## 4.2 Soil moisture, canopy interception and evapotranspiration

According to the majority of studies, tree growth in permafrost areas is limited by summer air temperatures and available water from snow melt, water accumulated within the soil in the previous year and permafrost thaw water (Kharuk et al., 2015; Sidorova et al., 2007). The amount of precipitation in the eastern Siberian Taiga is characteristically small compared to other areas, therefore it is expected that permafrost plays an important role in the existence of these forests (Sugimoto et al., 2002). Sugimoto et al. (2002) found that plants used rainwater during wet summers, but melt water from permafrost during drought summers. This indicates that permafrost provides the direct source of water for plants in drought summers and retains surplus water in the soil until the next summer. They conclude that if this system is disturbed by future warming, the forest stands might be seriously damaged in severe drought summers (Sugimoto et al., 2002). Our grassland site, which was supposedly forested until the 1950s, has dried up and has a much smaller organic layer and a maximum active layer thickness of $2.30\,\mathrm{m}$. The volumetric water content in forest soil is 10-20% higher despite the same amount of precipitation and a higher evaporative flux during the growing season. This points to the conclusion that permafrost plays an important role in regulating the hydrological conditions in this boreal forest area by holding the water table close to the surface which improves plant water supply.

## 4.3 Insulating litter and moss layer

The existence of a thick moss and organic layer on the forest ground can significantly lower ground temperatures due to the high insulation impact (Bonan and Shugart, 1989). The low bulk density and low thermal conductivity of the organic mat effectively insulate the mineral soil which causes lower soil temperatures and maintains a high permafrost table. A thick moss-organic layer on the forest floor is an important structural component of the forests-permafrost relationship, controlling energy flow, nutrient cycling, water relations, and through these, stand productivity and dynamics (Bonan and Shugart, 1989). At our forest site the moss coverage was found to range between 0 and 40% with thicknesses ranging from $0.005$ to $0.02\,\mathrm{m}$. This is a comparably thin moss layer but it was taken into account in the ground set-up for the forest site.

## 4.4 Snow-pack dynamics

Snow cover dynamics on the other hand seem to be a highly important factor in regard to the thermal evolution of the underlying permafrost ground (Gouttevin et al., 2012). Snow cover is an essential ecosystem component, acting as a radiation shield, insulator and a seasonal water reservoir. The forest canopy exerts a strong control on snow accumulation due to interception and reduced wind speeds (Price, 1988). To analyze characteristics in the snow cover evolution, we compare modeled and measured snow depths at the AWS (grassland site) with the modeled snow depth at the forest site. Snow depth modeled in grassland agrees well with the measured snow depth and reaches a maximum value of $0.26\,\mathrm{m}$ in late April (see Fig. 7).

Towards spring, the snow-pack in the forest accumulates to $1.2\,\mathrm{m}$. Snow melting starts around the same period but lasts longer up until the end of May. The snow-pack at the forest site exhibits different characteristics than at the grassland site. The found differences are clearly induced by the canopy structure controlling snow interception within the canopy, mass unload from the branches to the ground, sublimation, and snow compaction.

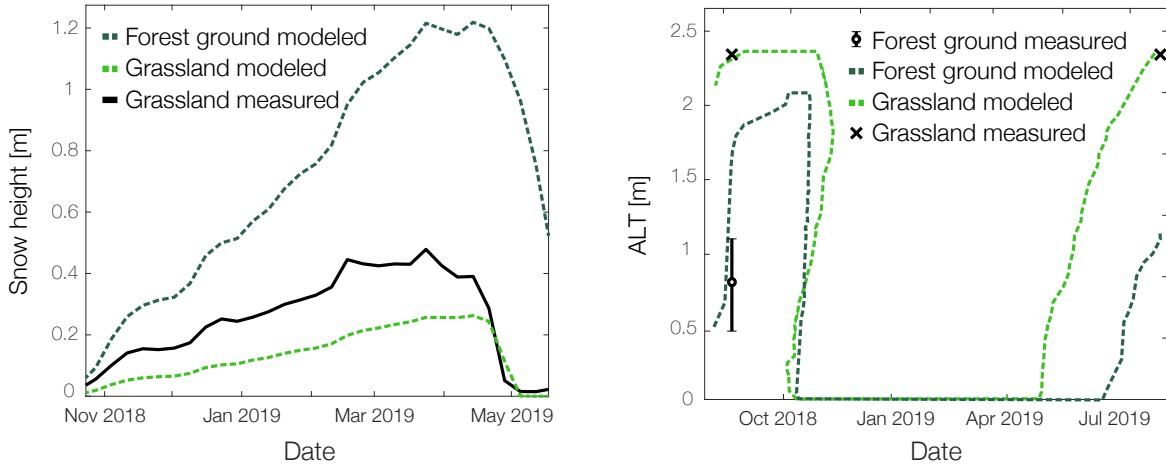

**Figure 7.** Left, Measured snow depth at AWS (black solid line) in the grassland, modeled snow depth in grassland (dashed light green) and modeled snow depth in forest (dashed dark green). Right, Active layer thickness (ALT) dynamics, measured ALT at the grassland and forest sites (black, point measurements in 2018 and 2019 (grassland only)), modeled ALT in forest (dashed light green) and modeled ALT in grassland (dashed dark green) sites. Top soil freezing starts at the beginning of October.

In addition, our simulations at the forest site show, an increase in longwave radiation, a decrease in solar radiation and the oppression of turbulent fluxes, which additionally lead to slower snow melting, less snow compaction, and therefore a higher snow-pack. This is in agreement with earlier work on snow-pack modeling in coniferous forests (Price, 1988). For example, Beer et al. (2007) note that vegetation effects such as solar radiation extinction and atmospheric turbulence have a far greater influence on snow cover dynamics in eastern Siberian boreal forests than snow interception alone. In addition, Grippa et al. (2005) found that leaf area index (LAI) and snow depth are highly connected.

To understand the high impact that coupling the vegetation has on the snow cover, we next study the surface energy balance simulated by our model for our specific forest study site and a hypothetical, sparse canopy with a LAI of $1\,\mathrm{m}^2\,\mathrm{m}^{-2}$, while keeping all other parameters the same (see Fig. 8). In snow-free periods the mean incoming solar radiation at the ground is 5 times higher in the sparse canopy simulation, which leads to a higher net radiation flux ($Q_{net}$). Turbulent fluxes ($Q_h$ and $Q_e$) are similar, which suggests that air circulation is blocked, even in a very sparse canopy. The high longwave radiation at the forest ground is persistent for the hypothetical sparse canopy as well, and longwave radiation remains the dominant energy component.

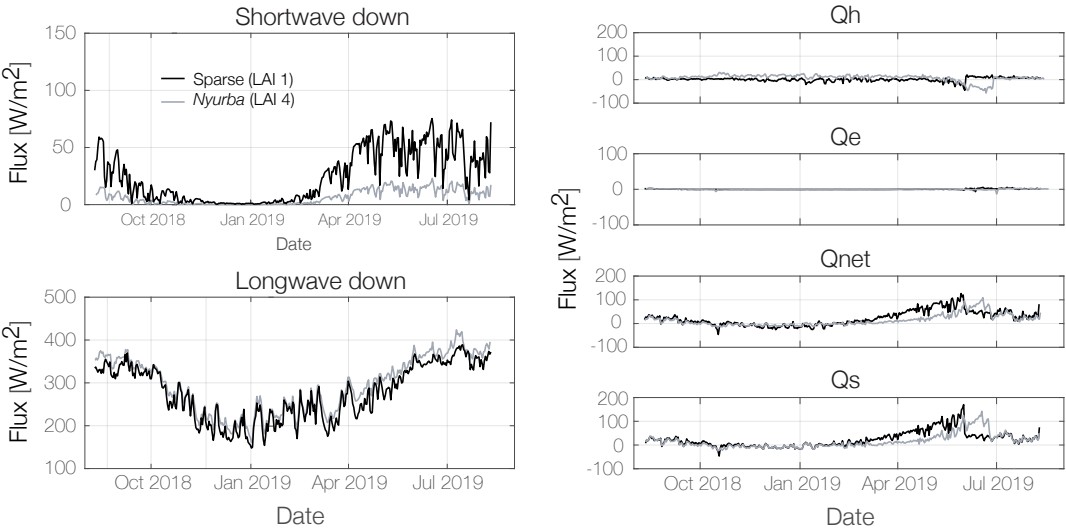

**Figure 8.** Left, modeled longwave and solar radiation at the forest ground for LAI $= 1\,\mathrm{m^2\,m^{-2}}$ (Sparse, black) and LAI $= 4\,\mathrm{m^2\,m^{-2}}$ (Nyurba, grey). Right, modeled turbulent fluxes ($Q_h$ and $Q_e$), net radiation ($Q_{net}$) and storage heat flux at the forest ground ($Q_s$).

Snow depth analysis further reveals that for a sparse forest canopy the maximum snow depth reaches $0.23\,\mathrm{m}$ only, resulting in a maximum ALT of $0.88\,\mathrm{m}$ and an annual average $\delta T$ of $0.8\,^{\circ}\mathrm{C}$. This confirms that the thermal differences between forest
and grassland sites are largely controlled by the impact of the canopy density on snow depth and density.

The snow-pack at our primary site (mixed forest) reaches a maximum thickness of $1.2\,\mathrm{m}$ which is in accordance with studies of boreal forest snow depths in other boreal regions such as in Canadian boreal regions, where i.e. Kershaw and McCulloch (2007) found varying mean snow-pack depths between $0.73\,\mathrm{m}$ and $1.3\,\mathrm{m}$ in different forest types and only $0.08\,\mathrm{m}$ in a tundra landscape. Further, Fortin et al. (2015) measured maximum snow-pack heights between $0.7\,\mathrm{m}$ and $0.9\,\mathrm{m}$ in a black spruce
dominated forest-tundra ecotone. Similar values were also found in a study in mixed boreal forests in Northeastern China (Chang et al., 2015) and in a more general large-scale approach for the circumpolar north (Zhang et al., 2018). This strong variability and heterogeneity in snow distribution has already been identified as a very important driver of the subsurface and hydrological regimes and runoff in unforested permafrost regions (Nitzbon et al., 2019).

Nevertheless, the strong delay between observed and modeled top ground thawing at the forest site (see Fig. 6) demands
further investigation. The snow compaction currently used in the snow module is dependent on wind speed only (see Sect. 2.3.3). Due to the coupled forest canopy, modeled wind speed at ground level is strongly reduced to the minimum value of $0.1\,\mathrm{m/s}$. Consequently, the snow compaction is remarkably low. To understand how much of the difference between modeled and measured spring GST can be explained by the underestimated snow compaction we simulate an extreme case, using the above canopy wind speed for the snow compaction processes (see Fig. 9). The use of the above-canopy wind speed and the
resulting high snow compaction reduces the difference between modeled and measured GST in spring by about 50%. This

reduction arises from the lower insulation capacity of the thinner snow-pack. However, the timing of top soil thawing does not improve. This may be explained by the fact that the snow cover has, on the one hand, a lower depth, but on the other hand, a higher density which results in the same snow water equivalent and an equally high amount of energy that is needed to melt the snow cover completely.

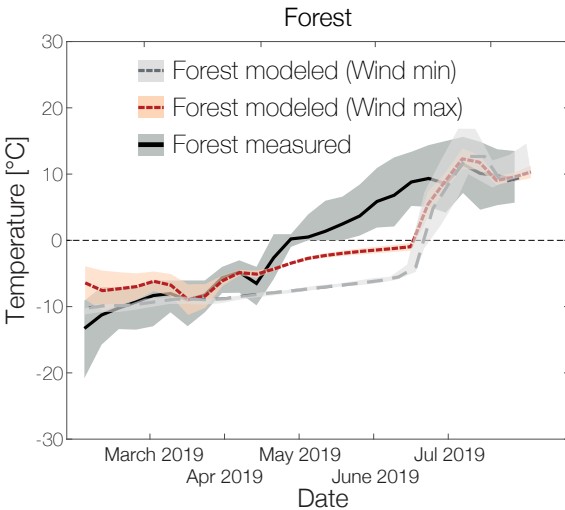

**Figure 9.** Modeled (Wind min (grey) and Wind max (red)) and measured (black) average weekly GST in $0.03\,\mathrm{m}$ depth in forest with standard deviation for modeled (light grey and light red) and measured (dark grey). Red represents a simulation using the wind speed at the top of the canopy as the input value for snow compaction. Grey shows the standard model simulation using the wind speed at the canopy bottom for the snow compaction mechanisms.

## 4.5 Applicability and model limitations

The presented model is largely able to reproduce recorded GSTs in forests. The detailed analysis of the annual cycle shows that the snow melt period in spring is biased at the forest and grassland sites. In reality, the ground warms up faster than modeled. In the forest this is most likely caused by a wrong representation of snow-pack compaction and melt. Our analysis reveals that an extreme case of snow compaction only partly reduces the difference between modeled and measured GST in spring. This points to more complex processes that control snow melt in forest than currently represented by the model. Thus, it would be highly desirable to obtain further field measurements in order to gain a better understanding of snow melt processes in boreal forests.

An aspect not represented in the model is the moisture transport and migration in frozen ground including the forming of ice lenses and excess ground ice, which can have a high impact on the local micro-topography and the surface energy balance. Furthermore, lateral water flow and snow redistribution may be important processes to be investigated in the future since they can strongly modify the ground thermal regime as well as the snow-pack development.

Additionally, more detailed field studies and modeling exercises on the variation of canopy densities and structures should be carried out in order to obtain a better understanding of the impact of dynamic forest stand development on permafrost and vice versa. In combination, the above model limitations could explain a great part of the described GST and ALT differences between measurements and modeled simulations.

To simulate the needle-tossing of deciduous larch, we have incorporated a leaf area index threshold between needle-tossing and leaf-out (10. October - 30. May) for simulations at our external validation site at Spasskaya-Pad (see Appendix C). This tunes the model towards a more detailed representation of larch-dominated forests, which are particular to the secondary study site and large parts of Eastern Siberia. The analysis reveals a satisfactory agreement between modeled and measured components of the surface energy balance below the predominantly deciduous forest canopy. The mixed forest cover at our primary study site only contains 7% of deciduous larch trees. The LAI reduction implemented is therefore very small and had no noticeable effect on our results. This modification can be used to study further taxa-specific interactions with permanently frozen ground. It would also be desirable to implement a spatially-explicit, dynamic vegetation model, such as the larch forest simulator (LAVESI, Kruse et al. (2016)) to further analyze the dynamic vegetation distribution under the recognition of the found interactions. This would allow us to simulate the vegetation response to changes in permafrost temperature and hydrology dynamically over a large timescale, and across a wide range of boreal forest ecosystems in Eastern Siberia.

## 5   Conclusions

This study presents a specific application of a novel, coupled multilayer forest-permafrost model which enables us to investigate the energy transfer and surface energy balance in permafrost underlain boreal forest of Eastern Siberia. By simulating interactions between the vegetation cover and permafrost, our modeling approach allows us to quantify and study the impact of the forest on the hydro-thermal regime of the permafrost ground below. An extensive comparison between measured and modeled energy balance variables (GST, $Q_e$, $Q_h$, $Q_{net}$, $S_{in}$ and $S_{out}$) reveals a satisfactory model performance justifying its application to investigate the thermal regime and surface energy balance in this complex ecosystem. Despite overall good performance, the field measurements reveal model shortcomings during the snow melt period. Based on this modeling exercise and field measurements, we investigate the thermal conditions of two landscape entities as they typically occur in the boreal zone. In regard to the forests insulation effect on permafrost and ongoing land cover transitions this study delivers important insights into the range of spatial differences and possible temporal changes that can be expected following landscape changes such as deforestation through fires, anthropogenic influences and afforestation in currently unforested grasslands or the densification of forested areas. The detailed vegetation model successfully calculates the canopy radiation and water budgets, leaf fluxes, as well as canopy turbulence and aerodynamic conductance. These canopy fluxes alter the below-canopy surface energy balance, the ground thermal conditions and the snow cover dynamics. We find a strong dampening effect of over $30\,^{\circ}\mathrm{C}$ on the annual ground surface temperature amplitude of the permafrost. Further, forested permafrost maintains a higher soil water content by controlling water storage in the ground. The forest cover alters the surface energy balance by inhibiting most of the solar radiation and suppressing turbulent heat fluxes. Additionally, we reveal that the canopy leads to a surplus in longwave

radiation trapped below the canopy, similar to a greenhouse. Therefore, and despite the canopy shading, the storage heat flux at the forest site is similar in magnitude to that simulated at the grassland site. In summary, we identify the following key points.

   i. The forest canopy effectively absorbs and reflects over 90% of incoming solar radiation, making canopy shading one of the main controlling mechanisms.

  ii. The vegetation cover suppresses the majority of the turbulent heat fluxes in the below-canopy space.

iii. The forest canopy enhances the longwave radiation below the canopy by up to 20%, similar to a greenhouse, which results in a comparable magnitude of storage heat flux for both, the forest and the grassland sites.

  iv. Forested permafrost holds a higher ground water content than the dry grassland site.

  v. Forest canopy shading leads to slower snow melting, less snow compaction, and therefore a higher snow-pack.

  vi. The differences in the thermal development of the forest and grassland sites are highly influenced by the depth, density
and the resulting insulation capacities of the snow cover, which is in turn controlled by the forest canopy density.

*Code and data availability.* The code is available at https://github.com/CryoGrid/CryoGrid/tree/vegetation. The iButton soil temperature data are available at https://doi.org/10.1594/PANGAEA.915174. The AWS data are available at: https://doi.org/10.1594/PANGAEA.919859. The high-resolution photogrammetric point clouds data used in Figure 1 are available at: https://doi.org/10.1594/PANGAEA.902259.

## Appendix A:  Bowen ratio and turbulent heat flux calculation

With the AWS equipped as a Bowen ratio station, $B$ is calculated following (Foken, 2016) as

$$B = \frac{c_p}{L_v} \times \frac{\Delta T}{\Delta q}, \tag{A1}$$

where the specific heat at constant pressure for moist air ($c_p$) is 1.006 kJ kg$^{-1}$K$^{-1}$ and the latent heat of vaporization of water ($L_v$) is 2260 kJ kg$^{-1}$. With $\Delta T$ being the temperature difference between the two air temperature sensors at heights $0.115\,\text{m}$ and $0.252\,\text{m}$. $\Delta q$ is the difference in specific humidity calculated from measured relative humidity ($\phi$), temperature and pressure.

Thereafter, latent heat flux ($Q_e$) is calculated as

$$Q_e = \frac{Q_s - Q_g}{1 + B}, \tag{A2}$$

and sensible heat flux ($Q_h$)

$$Q_h = (Q_s - Q_g)\frac{B}{1 + B}, \tag{A3}$$

with the storage heat flux ($Q_s$)

$$Q_s = L_{in} + S_{in} - L_{out} - S_{out}, \tag{A4}$$

and $Q_g$ as the convective ground heat flux.

## Appendix B: Direct and diffuse solar radiation components from cloud cover data

ERA interim cloud cover data (N) allows us to use a simple approach to differentiate the incoming shortwave radiation ($S_{in}$) into diffuse

$$S_{\text{in}_{\text{diffuse}}} = S_{in} * (0.3 + 0.7 * (N/8)^2), \tag{B1}$$

and direct,

$$S_{\text{in}_{\text{direct}}} = S_{in} - S_{\text{in}_{\text{diffuse}}}, \tag{B2}$$

components, based on Younes and Muneer (2007).

## Appendix C:  External validation site "Spasskaya-Pad"

Further validation of the model performance is performed for a well-studied research site in Spasskaya-Pad at N 62.14°, E 129.37°. This additional validation site is located $581\,\mathrm{km}$ from our primary study site, but allows to validate further model variables due to additional observational data. Spasskaya-Pad is a continuous permafrost region and the active layer depth is about $1.2\,\mathrm{m}$ in larch-dominated forests. Main tree species is Dahurian larch (*Larix gmelinii*) with a stand density of $840\,\mathrm{trees/ha}$. Understory vegetation (*Vaccinium*) is dense and $0.05\,\mathrm{m}$ high. In 1996 a $32\,\mathrm{m}$ observation tower was installed (Ohta et al., 2001) in larch dominated forest. Through the Arctic Data Archive system (ADS, https://ads.nipr.ac.jp/) we have been provided with the most recent, available meteorological and radiation data from beneath and above the larch-dominated forest canopy for the time-period 2017-2018 (Maximov et al., 2019). Each variable used here is measured at exactly 5-min intervals, except radiation (1-min). Ventilated shelters cover air temperature and humidity sensors. Net all-wave radiation and the four components of radiation are measured every minute, and the data loggers record average, maximum and minimum values. Upward and downward long-wave radiation is corrected using the sensed temperature at domes and sensor bodies. Ground temperature is measured at seven depths, and soil moisture at five depths. A more detailed description of the sensors can be found in Table 1 in Ohta et al. (2001). We have set-up and ran a 6-year simulation for this study site (2013-2019), using ERA-interim forcing data using the grid cell closest to the coordinate N 62.14°, E 129.37°. The measurement tower is situated in larch-dominated forest so that a simple leaf-off parameterisation is implemented. Following Ohta et al. (2001) we define a partial leaf-off period from 10. October - 30. May, resulting in a reduced winter LAI of $0.5\,\mathrm{m^2\,m^{-2}}$. Summer LAI is set to constant value of $1.9\,\mathrm{m^2\,m^{-2}}$ and we use the measured average tree height of $18\,\mathrm{m}$ for setting up the canopy structure. In order to ensure consistent model validation with the primary study site we used identical soil parameters for the external study site. All soil parameters used are summarized in Table A3 while Table A4 summarizes the constants used. We make use of canopy parameters defined by the PFT deciduous needleleaf due to the dominance of deciduous larch. The subsurface (soil) stratigraphy extends to $100\,\mathrm{m}$ below surface where the geothermal heat flux is set to a standard value of $0.05\,\mathrm{W\,m^{-2}}$ (Langer et al., 2011b). The ground is divided into separate layers in the model. The uppermost $8\,\mathrm{m}$ have a layer thickness of $0.05\,\mathrm{m}$, followed by $0.1\,\mathrm{m}$ for the next $20\,\mathrm{m}$, $0.5\,\mathrm{m}$ up to $50\,\mathrm{m}$ and $1\,\mathrm{m}$ thereafter. All remaining model parameters were set to default values as defined in previous studies (see Table A2) (Langer et al., 2011a, b, 2016; Westermann et al., 2016; Nitzbon et al., 2019, 2020). Similar to the primary study site we use ground surface temperature (GST) as one of the target variables for model validation, measured and modeled at $0.2\,\mathrm{m}$. In addition we use air temperature below the canopy, measured at the height of $1.2\,\mathrm{m}$, net radiation ($Q_{net}$), latent ($Q_e$) and sensible ($Q_h$) heat flux, and incoming ($S_{in}$) and outgoing ($S_{out}$) shortwave radiation flux at the ground surface as additional target variables allowing a comprehensive validation of the modeled heat and moisture exchange processes within and below the canopy.

We assess the surface energy balance by comparing the median weekly values of modeled and measured net radiation ($Q_{net}$), sensible heat flux ($Q_h$), latent heat flux ($Q_e$), and incoming ($S_{in}$) and outgoing ($S_{out}$) solar radiation at the forested site (see Fig. C1).

Modeled turbulent fluxes below the canopy are small during the snow-covered period and measurement data are not available during this period. Modeled and measured sensible heat flux in the snow-free period differ by $0.1\,\mathrm{W\,m^{-2}}$ only. Modeled latent heat flux is only a fourth of the measured value and therefore underestimated in our model. Modeled net radiation in snow-free period ($25.7\,\mathrm{W\,m^{-2}}$) is slightly above measured net radiation ($19.4\,\mathrm{W\,m^{-2}}$). For snow-covered period median modeled net radiation is slightly below the measured median value. The incoming shortwave radiation measured and modeled for the forest site fit well with differences well below $10\,\mathrm{W\,m^{-2}}$. The standard deviation of measured values is higher for all variables except snow-covered net radiation.

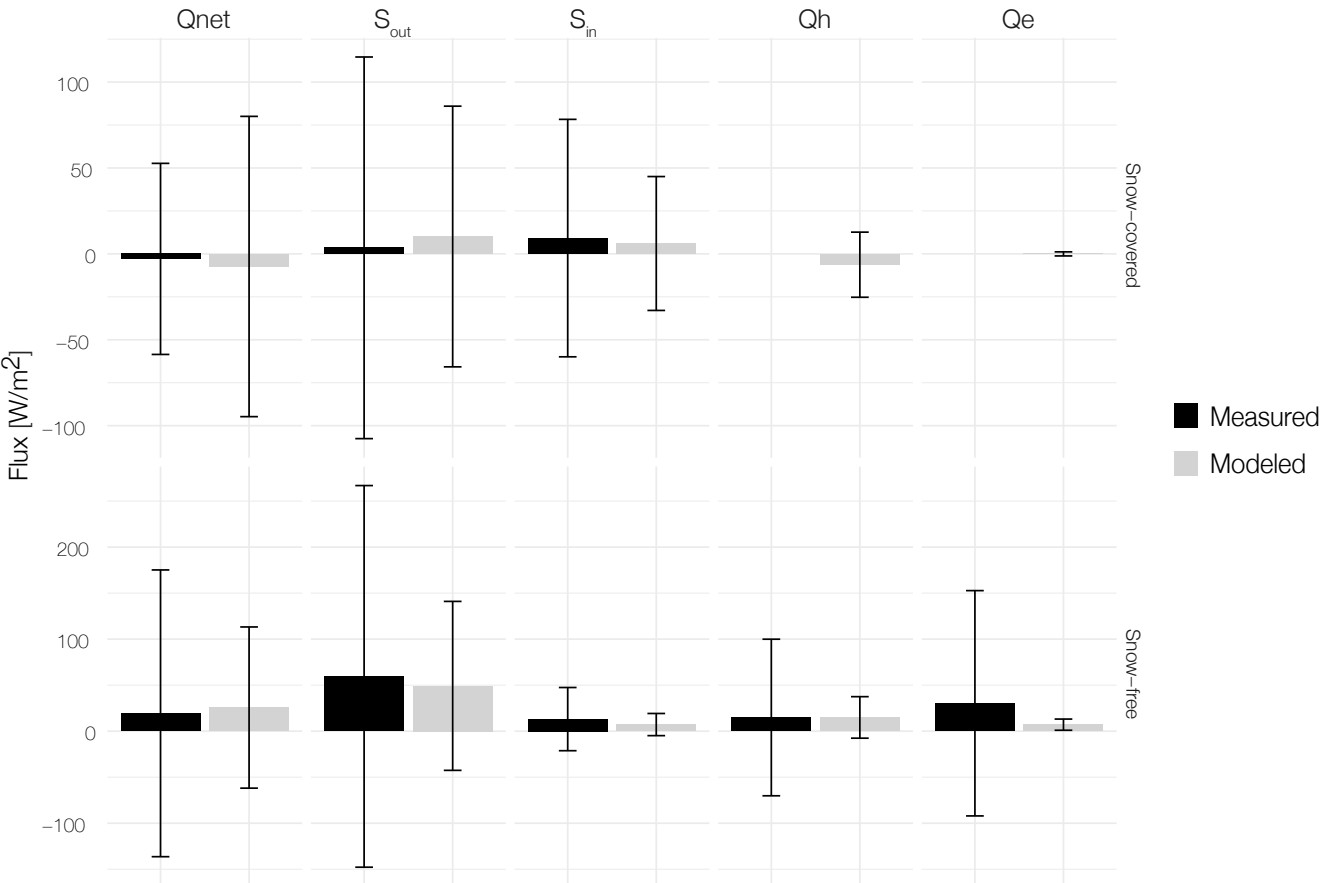

**Figure C1.** Modeled (grey) incoming and outgoing solar radiation ($S_{in}$, $S_{out}$) and turbulent fluxes ($Q_h$, $Q_e$, $Q_{net}$) for snow-covered (28.10.2017-27.04.2018, above) and snow-free (10.10.2017-27.10.2017 and 28.04.2018-10.10.2018, below) periods at the ground surface of forest. The bars indicate median values while the whiskers show the corresponding standard deviations.

In a second step, we compare the modeled and measured annual, snow-free and snow-covered median GST and air temperature below the canopy to understand the overall model performance regarding the thermal regime of the surface and the ground and the relative temperature differences between the model and measurements (see Fig. C2).

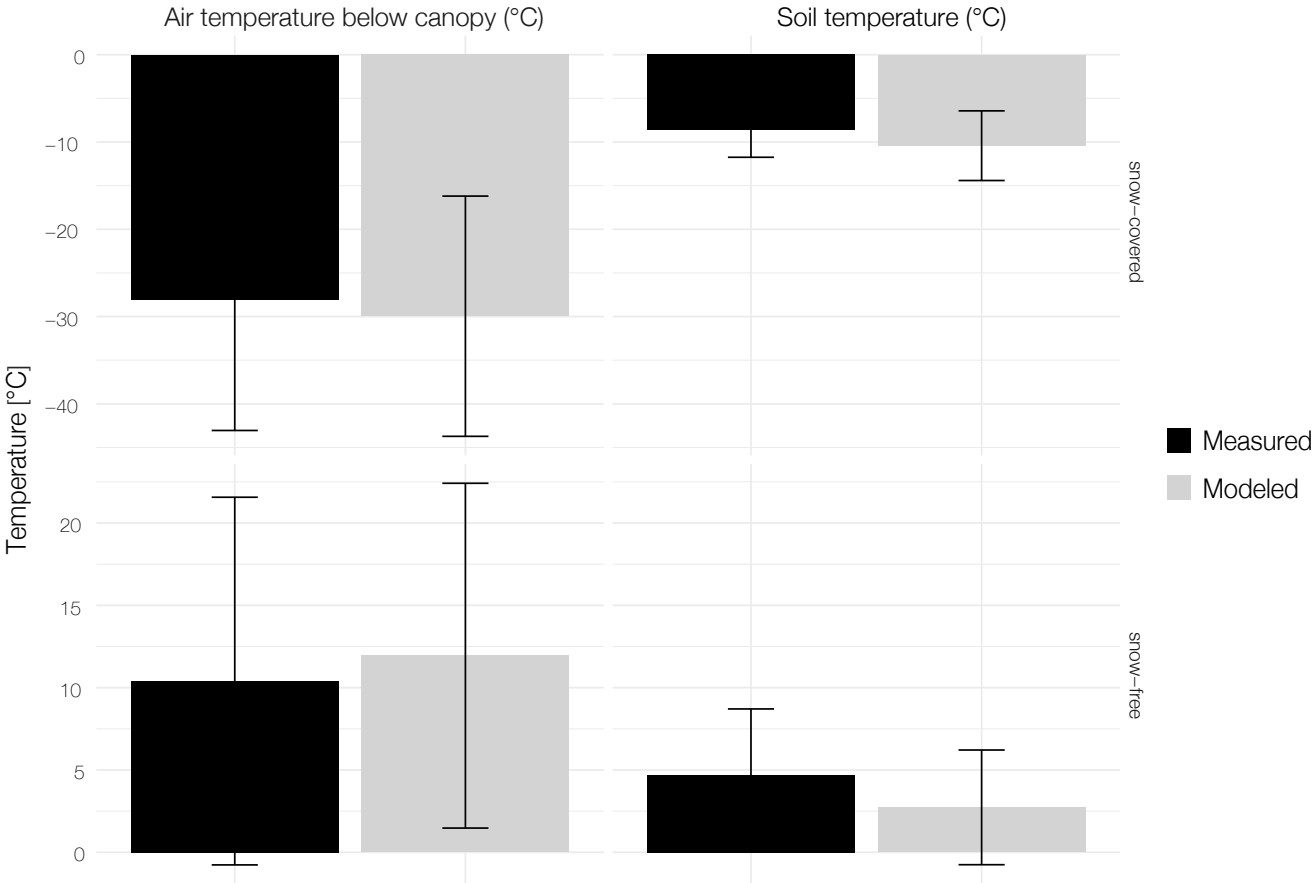

**Figure C2.** Left, modeled (grey) and measured (black) air temperature (°C) below the canopy. Right, ground surface temperature (°C). Both for snow-covered (28.10.2017-27.04.2018, above) and snow-free (10.10.2017-27.10.2017 and 28.04.2018-10.10.2018, below) periods at the forest site in Spasskaya-Pad. The bars indicate median values while the whiskers show the corresponding standard deviations.

The highest deviation between modeled and measured temperatures is found in the GST of the snow-free period. Here, the model shows a cold bias of $-2\,°$C. For the snow-covered period the difference is $1.8\,°$C. For the air temperature below the canopy the difference between modeled and measured in the snow-free period is $1.5\,°$C, for snow-covered the difference is again $1.8\,°$C. This falls into the range of $1.5-2\,°$C that are commonly used for validation purposes (Langer et al., 2013; Westermann et al., 2016).

Overall our analysis reveals a satisfactory agreement between modeled and measured components of the surface energy balance below the canopy. Thus, we argue that the performance of the model at the external study site justifies its application at the primary study site in Nyurba where below canopy fluxes were not acquired.

**Table A1.** Sensors used for field measurements.

| Sensor | Brand | Measurement | Accuracy |
|---|---|---|---|
| Temperature and relative humidity probe (HMP155A) | Vaisala | Air temp. / rel. Humidity | $\pm 1\%$ ($15 - 25\,^\circ$C) |
| Alpine wind monitor (05103-45) | R. M. Young C. | Wind speed / direction | 1% of reading |
| Sonic ranging sensor (SR50A) | Campbell | Snow depth | 0.4% of height |
| Barometric sensor (CS100) | Setra | Barometric pressure | $\pm 0.5$ mb ($20\,^\circ$C) |
| 4-Component Net Radiometer (NR01) | Hukseflux | S/L in and out | 10% daily totals |
| Thermistor probe (107) | Campbell | Soil temperature | $0.2\,^\circ$C |
| Heat flux sensor (HFP01) | Hukseflux | Ground heat flux | $\pm 3\%$ |
| Raingauge tipping bucket unheated (52203) | R. M. Young C. | Precip. (liquid) | 2% up to 25 mm hr$^{-1}$ |
| Water Content Reflectometer (CS616) | Campbell | Soil moisture | $\pm 2.5\%$ VWC |
| Hobo 4 Channel Data Logger + Temperaturesensor | Onset | Soil temperature | $\pm 2$ mV $\pm 2.5\%$ abs. reading |
| iButton (DS1922L) | Maxim Integrated | Soil temperature | $\pm 0.5\,^\circ$C ($-10 - 65\,^\circ$C) |

**Table A2.** Overview of the CryoGrid parameters used

| Process / Parameter | | Value | Unit | Source |
|---|---|---|---|---|
| Density falling snow | $\rho_{snow}$ | 300 | kg m$^{-3}$ | *Kershaw and McCulloch (2007)* |
| Albedo ground | $\alpha$ | 0.3 | - | *field measurement* |
| Roughness length | $z_0$ | 0.001 | m | *Westermann et al. (2016)* |
| Roughness length snow | $z_{0snow}$ | 0.0001 | m | *Boike et al. (2019)* |
| Geothermal heat flux | $F_{lb}$ | 0.05 | W m$^{-2}$ | *Westermann et al. (2016)* |
| Thermal conductivity mineral soil fraction | $k_{mineral}$ | 3.0 | W m$^{-1}$ K$^{-1}$ | *Westermann et al. (2016)* |
| Emissivity | $\epsilon$ | 0.99 | - | *Langer et al. (2011a)* |
| Root depth | $D_T$ | 0.2 | m | *field measurement* |
| Evaporation depth | $D_E$ | 0.1 | m | *Nitzbon et al. (2019)* |
| Hydraulic conductivity | $K$ | 10$^{-5}$ | m s$^{-1}$ | *Boike et al. (2019)* |

**Table A3.** Ground set-up for simulations. Depth in [m], all other in volumetric fractions (unitless)

|  | Top depth | Water/Ice | Mineral | Organic | Field capacity | Natural porosity |
|---|---|---|---|---|---|---|
| Forest | 0 | 0.6 | 0 | 0.2 | 0.5 | 0.8 |
|  | 0.08 | 0.6 | 0.1 | 0.2 | 0.5 | 0.7 |
|  | 0.16 | 0.6 | 0.4 | 0 | 0.5 | 0.6 |
| Grassland | 0 | 0.5 | 0.4 | 0.1 | 0.5 | 0.5 |
|  | 0.04 | 0.4 | 0.6 | 0 | 0.5 | 0.4 |
|  | 0.1 | 0.4 | 0.6 | 0 | 0.5 | 0.4 |

**Table A4.** Constants

| Constants | Value | Unit |
|---|---|---|
| von Karman | 0.4 | - |
| Freezing point water (normal pres.) | 273.15 | K |
| Latent heat of vaporization | 2.501 x 10^6 | J kg$^{-1}$ |
| Molecular mass of water | 18.016/1000 | kg mol$^{-1}$ |
| Molecular mass of dry air | 28.966/1000 | kg mol$^{-1}$ |
| Specific heat dry air (const. pres.) | 1004.64 | J kg$^{-1}$ K$^{-1}$ |
| Density of fresh water | 1000 | kg m$^{-3}$ |
| Density of ice | 917 | kg m$^{-3}$ |
| Heat of fusion for water at $0\,^\circ$C | 0.334 x 10^6 | J kg$^{-1}$ |
| Thermal conductivity of water | 0.57 | m$^{-1}$ K$^{-1}$ |
| Thermal conductivity of ice | 2.29 | W m$^{-1}$ K$^{-1}$ |
| Kinem. visc. air ($0\,^\circ$C, 1013.25 hPa) | 0.0000133 | m$^2$ s$^{-1}$ |
| Sp. heat water vapor (const. pr.) | 1810 | J kg$^{-1}$ K$^{-1}$ |

**Table A5.** Multilayer canopy parameters

| Parameter PFT NET boreal | Value | Unit | Source |
|---|---|---|---|
| Leaf angle dep. from spherical | 0.01 | - | Bonan (2002) |
| Leaf reflectance (VIS/NIR) | 0.07/0.35 | - | Bonan (2002) |
| Stem reflectance (VIS/NIR) | 0.16/0.39 | - | Bonan (2002) |
| Leaf transmittance (VIS/NIR) | 0.05/0.01 | - | Bonan (2002) |
| Stem transmittance (VIS/NIR) | 0.001/0.001 | - | Bonan (2002) |
| Maximum carboxylation rate ($25\,°C$) | 43 | umol m$^{-2}$ s$^{-1}$ | Bonan (2002) |
| Photosynthetic pathway | C3 | - | Bonan (2002) |
| Leaf emissivity | 0.98 | - | Bonan (2002) |
| Quantum efficiency a | 0.06 | umol $CO_2$ umol photon$^{-1}$ | Bonan (2002) |
| Slope m | 6 | - | Bonan (2002) |
| Leaf dimension | 0.04 | m | Bonan (2002) |
| Roughness length | 0.055 | m | Bonan (2002) |
| Displacement height | 0.67 | m | Bonan (2002) |
| Root distribution (a/b) | 7.0/2.0 | - | Bonan (2002) |
| Min. vapor pressure deficit | 100 | Pa | Bonan (2019) |
| Plant capacitance | 2500 | mmol $H_2O$ m$^{-2}$ leaf area MPa$^{-1}$ | Bonan (2019) |
| Minimum leaf water potential | -2 | MPa | Bonan (2019) |
| Stem hydraulic conductance | 4 | mmol $H_2O$ m$^{-2}$ leaf area s$^{-1}$ MPa$^{-1}$ | Bonan (2019) |
| Atmospheric $CO_2$ | 380 | umol mol$^{-1}$ | Bonan (2019) |
| Atmospheric $O_2$ | 209 | mmol mol$^{-1}$ | Bonan (2019) |
| Soil evaporative resistance | 3361.509 | s m$^{-1}$ | Bonan (2019) |
| Specific heat of dry-wet soil | 1396 | J kg$^{-1}$ K$^{-1}$ | Oleson et al. (2013) |
| Specific heat of fresh $H_2O$ | 4188 | J kg$^{-1}$ K$^{-1}$ | Oleson et al. (2013) |
| Specific leaf area at top of canopy | 0.01 | m$^2$ g$^{-1}$ C | Bonan et al. (2018) |
| Fine root biomass | 500 | g biomass m$^{-2}$ | Bonan (2019) |
| Leaf drag coefficient | 0.25 | - | Bonan (2019) |
| Foliage clumping index | 0.7 | - | Bonan (2019) |

**Table A6.** Further ground parameters needed by the vegetation

| Soil parameters (sandy clay loam) | Value | Unit | Source |
|---|---|---|---|
| Soil layer Clapp Hornberger "b" | 4.05 / 4.38 / 10.4 | - | Bonan (2019) |
| Alpha (empirical param.) | 0.059 | $cm^{-3}$ | Bonan (2019) |
| n (pore size distr. index) | 1.48 | - | Bonan (2019) |
| initial Porosity | 0.8 / 0.7 / 0.6 | $m^3\ m^{-3}$ | Bonan (2019) |
| Soil layer depth / thickness | 0.1 / 0.1 / 0.7 | m | - |
| Interface depth | 0.05 / 0.15 / 0.45 | m | - |
| Number of soil layers | 3 | - | - |

*Author contributions.* SMS designed the study, developed and implemented the numerical model, carried out and analysed the simulations, prepared the results figures and led the paper preparation. ML, SW, JB, UH and SK co-designed the study, and interpreted the results. SMS, ML, and TSvD implemented the code in the model and designed the model simulations. SMS, WC, LP, EZ, UH and SK prepared and conducted the field work in 2018, SMS and EZ conducted the field work in 2019. SMS wrote the paper with contributions from all co-authors. UH, ML and JB secured funding.

*Competing interests.* No competing interests are present

*Disclaimer.* TEXT

*Acknowledgements.* SMS is thankful to the POLMAR graduate school, the Geo.X Young Academy and the WiNS program at the Humboldt University of Berlin for providing a supportive framework for her PhD project and helpful courses on scientific writing and project management. Further, SMS is very grateful for the help during fieldwork in 2018 and 2019, especially for the help from Levina Sardana Niko-
575 laevna, Alexey Nikolajewitsch, Lena Ushnizkaya, Luise Schulte, Frederic Brieger, Stuart Vyse, Elisbeth Dietze, Nadine Bernhard, Boris K. Biskaborn, Iuliia Shevtsova, as well as my co-authors Luidmila Pestryakova and Evgeniy Zakharov. Additionally, SMS would like to thank Stephan Jacobi, Alexander Oehme, Niko Borneman, Peter Schreiber and my co-author William Cable for their help in preparing for field work and the entire PermaRisk and Sparc research groups for their ongoing support. Finally, SMS would like to thank the editor, Alexey V. Eliseev, and two anonymous reviewers for their comments and suggestions which have greatly improved our manuscript.
This study has been supported by the ERC consolidator grant Glacial Legacy of Ulrike Herzschuh (grant no. 772852). Further, the work was supported by the Federal Ministry of Education and Research (BMBF) of Germany through a grant to Moritz Langer (no. 01LN1709A). Funding was additionally provided by the Helmholtz Association in the framework of MOSES (Modular Observation Solutions for Earth Systems). LP was supported by the Russian Foundation for Basic Research (grant no. 18-45-140053 r_a), Ministry of Science and Higher

Education of the Russian Federation (grant no. FSRG-2020-0019). SW acknowledges funding by Permafost4Life (Research Council of Norway, grant no. 301639).

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
