# Peer review of "Variability of the Surface Energy Balance in Permafrost Underlain Boreal Forest"

_Biogeosciences, 2020_

## Referee Comment (RC1) · Anonymous Referee #1 · 28 Jul 2020

Stuenzi et al. use measurements and modelling of surface energy balance processes for a boreal forest and grassland site in Siberia to explore the impact of vegetation characteristics on ground thermal dynamics and snow cover. They couple a 1-D land surface model to a multilayer canopy model to account for radiative transfer through the canopy and for soil thaw dynamics. They find that the forest canopy efficiently reduces solar radiation at the forest floor causing slower soil warming and thaw and delayed snow melt.

The manuscript aims to better describe how vegetation interacts with soil thermal regimes in the continuous permafrost zone of Siberia. The authors provide an important and detailed description of the most relevant vegetation-ground interactions, which can help to better understand permafrost responses in a warming climate. One

concern is that no radiation, snow, or Bowen ratio measurements were available at the forest site. I know of the difficulties of setting up long-term observations in such environments, but the performance of the model simulations for the forest site - which is the major focus of this paper – cannot be properly assessed without these observations. The authors discuss this shortcoming in the discussion, but – in my opinion – they need to justify better then why the modelling results should be trusted. The modelled forest GST seem to reasonably fit the measured GST, but modelled soil thaw is delayed compared to observations. This raises the question if the modelling results regarding snow phenology are meaningful.

The manuscript provides a good overview and description of the relevant surface energy balance processes, but the authors could highlight how their study advances our understanding of surface energy balance processes in the forested permafrost zone. Bonan and Shugart (1989) outlined many of the relevant interactions and, for example, Chasmer et al. (2011, DOI: 10.1002/ppp.724) present results on forest canopy effects on radiative processes in a permafrost environment.

Other comments

Page 2, Line 19: In some cases, permafrost thaw and forest loss can also lead to increased $CO_2$ uptake as shown for thawing ice-rich permafrost in northwestern Canada and Alaska.

Page 3, Line 25: By how much has the summer precipitation decreased?

Page 5, Line 20: How was tree height estimated? Are there any ground-based LAI measurements in similar forest types?

Table 1: Perhaps, the equations in Table 1 could be shown in the table itself or at least qualitatively described. As it is now, the content of the table is not easy to grasp.

Page 6, Line 4: Why did the authors choose the PFT "deciduous needleleaf forest"? They mention that the site is dominated by Picea obovata (92%). Wouldn't an evergreen needleleaf parameterisation be more adequate? Also, if using the PFT "deciduous needleleaf forest", wouldn't it be necessary to include a phenology module? This is partly discussed later in the manuscript, but should be already mentioned here.

Page 8, line 8: How was soil thermal conductivity parameterised?

Figure 3: What are the error bars showing? What is the input to calculate error bars (hourly, daily, weekly data)? It seems as if the variability (i.e. error bars) for the observed grassland turbulent energy fluxes is much larger than the modelled variability.

Table 2: Is this the same data as shown in Figure 5? If so, the authors could think about showing only one of them (figure or table).

Figure 6: Most of the model-observation comparison is of qualitative nature. The authors could add some performance metrics (e.g., RMSE, R2, bias. . .)

Page 18, line 12: The authors report a bias in modelled GST during the winter. Since GST measurements are available, the forest-grassland comparison could be more meaningful if it were based on observational data.

Page 22, line 17: Could lateral flow of water contribute to differences in ground water content?

---

## Referee Comment (RC2) · Anonymous Referee #2 · 19 Aug 2020

Comments on "Variability of the surface energy balance in permafrost underlain boreal forest" submitted by Simone Maria Stuenzi et al. to Biogeosciences

In this manuscript, the authors developed a biophysical atmosphere-land interaction model (CryoGrid) and examined its performance by comparing with observations at a mixed forest in East Siberia. By coupling with a multi-layer canopy scheme (CLM-ml v0), they accounted for the effects of vegetation canopy on surface processes. They conducted a series of simulations with different land covers (forest and grassland) and different leaf area index or canopy density. They found that coverage with high leaf area index affects surface energy budget, such as solar radiation transfer and insulation, and that it could play important roles in snow and permafrost dynamics such as active layer thickness.

[Figure]

**BGD**

I agree that terrestrial ecosystems in northern high latitude is remarkable in terms of climate change, especially permafrost thawing that is thought as one of the tipping elements of the Earth system. Then, developing advanced models simulating boreal ecosystem processes is highly important. Nevertheless, I could not find out what is the original contribution of this study. For example, the model developed in this study seems similar to previous models such as CLM, ORHIDEE, LPJ etc. These previous models have already implemented leaf phenology and dynamic vegetation; at least some versions include permafrost dynamics.

Another concern on this study is that the authors used a very limited amount of observations. Especially, they validated the model performance to capture temporal variability only for surface temperature. Because many flux measurement sites are operating and providing a variety of observational data, I highly recommend validating the model by using a larger number of biophysical variables including energy and water fluxes. Also, I have a concern about the overly simplified simulation conditions, such as the lack of seasonal change in leaf area index in forest. Though in situ data were not available, at present, we can obtain time-series of leaf area index from satellite remote sensing as a good proxy.

Based on the inadequacy of scientific insights and model simulations, I cannot recommend this manuscript to accept for publication in the present form. The manuscript is too descriptive and needs more model validation with observations. Also, the authors need to devise model simulations to make insightful discussions.

Minor points

Page 3 Section 2.1: Can you give information on vegetation conditions, such as leaf area index and tree density?

Page 7 Line 3: I am not sure how within-canopy wind profile was parameterized and simulated by the multi-layer canopy model. Can you explain briefly, because this is an important feature of multi-layer canopy models?

Page 7 Line 7: Does the snow module include water loss by sublimation in winter? Under frigid environment like east Siberia, sublimation may not be negligible for snow water budget. Page 13 Figure 3: Can you give units to y-axis? Maybe, in W m–2.

Page 15 Table 2: Can you give other statistic metrics such as root-mean square error between measured and modeled values?

Page 17 Figure 7: Even in forest site, it is difficult for me to imagine 1m snow depth in eastern Siberia, where typical annual precipitation is 200–300 mm. Can you give observed annual precipitation at the study site?

Page 20 Section 4.3: This section looks to be a summary of Bonan and Shugart (1989). What is the key message of the section derived from the present study?
* * *

---

## Author Comment (AC1) · 19 Sep 2020

Stuenzi et al. use measurements and modelling of surface energy balance processes for a boreal forest and grassland site in Siberia to explore the impact of vegetation characteristics on ground thermal dynamics and snow cover. They couple a 1-D land surface model to a multilayer canopy model to account for radiative transfer through the canopy and for soil thaw dynamics. They find that the forest canopy efficiently reduces solar radiation at the forest floor causing slower soil warming and thaw and delayed snow melt.

The manuscript aims to better describe how vegetation interacts with soil thermal regimes in the continuous permafrost zone of Siberia. The authors provide an important and detailed description of the most relevant vegetation-ground interactions, which can help to better understand permafrost responses in a warming climate. One concern is that no radiation, snow, or Bowen ratio measurements were available at the forest site. I know of the difficulties of setting up long-term observations in such environments, but the performance of the model simulations for the forest site - which is the major focus of this paper – cannot be properly assessed without these observations.

The authors discuss this shortcoming in the discussion, but – in my opinion – they need to justify better then why the modelling results should be trusted. The modelled forest GST seem to reasonably fit the measured GST, but modelled soil thaw is delayed compared to observations. This raises the question if the modelling results regarding snow phenology are meaningful.

The manuscript provides a good overview and description of the relevant surface energy balance processes, but the authors could highlight how their study advances our understanding of surface energy balance processes in the forested permafrost zone. Bonan and Shugart (1989) outlined many of the relevant interactions and, for example, Chasmer et al. (2011, DOI: 10.1002/ppp.724) present results on forest canopy effects on radiative processes in a permafrost environment.

*We thank the reviewer for this positive overall evaluation of our study and for taking the time to review our manuscript. We have worked through all of the posed questions and suggestions made by the reviewer, which has improved our manuscript. Please note that any changes and additions to the text that we propose for the revised manuscript are highlighted in* **bold.**

*With this manuscript we, indeed, aim at improving our understanding of how the vegetation interacts with the solid thermal regime in the*

*continuous permafrost zone of eastern Siberia. We would like to thank the reviewer for acknowledging the importance of this detailed description of some of the most relevant vegetation-ground interactions in understanding permafrost responses to a warming climate.*

*Indeed, setting up such long-term measurements in these environments is highly challenging, but we agree that the model performance cannot be assessed completely without further integration of measurements. The model does successfully reproduce the measured ground surface temperatures (GST) of the monitored year (August 2018 - August 2019). Nevertheless, GST does not provide a full picture of the surface energy balance. Therefore, we suggest to further validate the model performance with additional measurements. Acquisition of sub-canopy radiation data is, as recognized by the reviewer, highly challenging and was unavailable within the scope of the fieldwork underlying the presented study. Since not many monitoring sites exist around Siberia, we suggest using existing and available data from the rather near, well-documented and well-studied research site Spasskaya-Pad at 62°14'N, 129°37'E. Through the Arctic Data Archive system (ADS) we have been provided meteorological and radiation data from beneath and above the larch-dominated forest canopy for 2018. This data can be used for additional model validation. We suggest adding this additional model validation to the appendix of our manuscript since it is a rather technical aspect, which is not directly related to our major study site.*

*Preliminary paragraph added to the Appendix:* **"For further validation of the model performance we use existing and available data from the rather near, well-documented and well-studied research site Spasskaya-Pad at 62°14'N, 129°37'E. Through the Arctic Data Archive system (ADS) we have been provided meteorological and radiation data from beneath and above the larch-dominated forest canopy for 2018. Therefore, we have set-up and ran a 5-year simulation for this study site, using ERA-interim forcing data for the coordinate above and a summer LAI of 3.66 $m^2m^{-2}$ (following the measurement-based LAI in Ohta et al. 2001) and a tree height of 18 m. Since the study site is larch-dominated we have now implemented a simple leaf-off parameterisation which is used here. Winter LAI is set to 1.66 $m^2m^{-2}$ (again based on Ohta et al. 2001) for the leaf-off period from 10. October - 10. April."**

*A detailed analysis of this validation run will be presented in the following revised manuscript. However, preliminary analyses of simulation results show a good fit with the modeled surface energy balance and justify the use of the model in the current version.*

*We further thank the reviewer for the suggested literature such as Bonan and Shugart (1989) who outlined relevant interactions and Chasmer et al. (2011) that present results on forest canopy effects on radiative processes in a permafrost environment. Bonan et al. (1989) is one of our main sources providing the overall framework of this study and is discussed on p.2, l.14 and on p.18, l.7 and l.10.*

*We have now carefully studied the article by Chasmer et al. (2011) which does present very interesting results for interaction*

*processes in a discontinuous permafrost zone in Canada. From this
study we have learned that vegetation on the edges of permafrost
plateaus tends towards reduced fractional canopy cover (by up to
50%) and reduced canopy heights (by 16-30%). The reduced biomass can
cause a positive feedback because of lower canopy shading (up to 1h
per day less), which leads to an increase in incident radiation at
the ground (+16% at open sites) and higher longwave radiation losses
(+74% at open plateau sites). We will incorporate this important
reference in the following sentence in the introduction (p.2, l.15):*
**"Changing climatic conditions can promote an increasing active layer
depth or trigger the partial disappearance of the near surface
permafrost. Further, extensive ecosystem shifts such as a change in
composition, density or the distribution of vegetation (Holtmeier
and Broll, 2005; Pearson et al., 2013; Gauthier et al., 2015; Kruse
et al., 2016; Ju and Masek, 2016) and resulting changes to the
below- and within-canopy radiation fluxes (Chasmer et al., 2011)
have already been reported."**

Other comments

Page 2, Line 19: In some cases, permafrost thaw and forest loss can also lead to increased

CO2 uptake as shown for thawing ice-rich permafrost in northwestern Canada and Alaska.

*We agree with the importance of this finding and suggest rearranging
the sentence to the following (p. 2, l. 19):* *"Changes to the
vegetation - permafrost dynamics can have a potentially high impact
on the numerous feedback mechanisms between the two ecosystem
components.* **Increased soil carbon release from thawing permafrost
through the delivery of soil organic matter to the active carbon
cycle (Schneider Von Deimling et al., 2012) is modified by
vegetation changes, which can compensate for carbon losses due to an
increased $CO_2$ uptake (as observed at ice-rich permafrost sites in
northwestern Canada and Alaska, Estop-Aragonés et al., 2018) or even
further accelerate total carbon loss (Romanovsky et al., 2017)"**

Page 3, Line 25: By how much has the summer precipitation decreased?

*We agree that this is an important question and we propose to add
the following information (p.3, l.25):* **"Annual precipitation showed
an increasing trend from 1900 until 1990, mainly due to an increase
in wintertime precipitation. Between 1995 and 2002, summertime
precipitation has decreased by -16.9 mm in August and -4.2 mm in
July (see table 1 in Hayasaka (2011) for further details)."**

Page 5, Line 20: How was tree height estimated? Are there any ground-based LAI
measurements in similar forest types?

*We thank the reviewer for this question and have added the following
information to the manuscript (p.5, l.20):* **"In a vegetation survey
along a 150 m transect from the grassland into the forest, the tree
height of every tree within a 2 m distance was estimated. Trees <2 m
were measured with a measuring tape, trees >2 m were measured with a
clinometer or visually estimated after repeated comparisons with
clinometer measurements."**

*Further, we recognize that the information given on LAI estimation on p.10, l.26 is insufficient, therefore we have modified the paragraph to the following, more detailed description:* **"LAI can be estimated from satellite data, calculated from below-canopy light measurements or by harvesting leaves and relating their mass to the the canopy diameter. Ohta et al. (2001) have described the monitored deciduous-needleleaf forest site at Spasskaya Pad research station, which has comparable climate conditions but is larch-dominated. The value of the tree plant area index (PAI), obtained from fish-eye imagery and confirmed by litter fall observations, varied between 3.71 $m^2m^{-2}$ in the foliated season and 1.71 $m^2m^{-2}$ in the leafless season. This value does not include the ground vegetation cover. Further, Chen et al. (2005) compared ground-based LAI measurements to MODIS values at an evergreen-dominated study area (57.3° N, 91.6° E) south-west of the region discussed here, around the city of Krasnoyarsk. The mixed forest consists of spruce, fir, pine and some occasional hardwood species (birch and aspen). They find LAI values between 2 $m^2m^{-2}$ and 7 $m^2m^{-2}$. To assess the LAI we use data from literature** *and the experience from the repeated field work at the described site.* **Following Kobayashi et al. (2010) who conducted an extensive study using satellite data, the average LAI for our forest type is set to 4 $m^2m^{-2}$ and stem area index (SAI) is set to 0.05 $m^2m^{-2}$, resulting in a plant area index (PAI)of 4.05 $m^2m^{-2}$ and 9 vegetation layers for model simulations."**

Table 1: Perhaps, the equations in Table 1 could be shown in the table itself or at least qualitatively described. As it is now, the content of the table is not easy to grasp.

*We agree with the reviewer that table 1 has little additional value.* **We have added Eq. 2 and Eq. 4 to the table directly.**

Page 6, Line 4: Why did the authors choose the PFT "deciduous needleleaf forest"? They mention that the site is dominated by Picea obovata (92%). Wouldn't an evergreen needleleaf parameterisation be more adequate? Also, if using the PFT "deciduous needleleaf forest", wouldn't it be necessary to include a phenology module? This is partly discussed later in the manuscript, but should be already mentioned here.

*We thank the reviewer for this remark. The PFT evergreen needleleaf (NET) is used for the simulations.* **This has been corrected in table A5 and on page 6.** *As discussed later (p.21, l.14), the development and implementation of a phenology module was out of scope for this study, mainly because of the little presence of deciduous taxa at the chosen study site.* **For a more detailed study of larch-dominated forest ecosystems, a simple phenology module has been implemented for the additional validation site at Spasskaya Pad where a much higher amount of deciduous taxa is present. This consists of a simple winter leaf-off parameterisation and results in a lower winter LAI (based on measurements from Ohta et al. 2001) and a leaf-off period from 10. October - 10. April (see the preliminary paragraph above describing the Spasskaya Pad validation site).**

Page 8, line 8: How was soil thermal conductivity parameterised?

*We agree that this information should be added to our manuscript and have done so on p.8, l.10:* **"Soil thermal conductivity is parameterised following Westermann et al. (2013 and 2016) and is based on the parameterization in Consenza et al. (2003). The thermal**

*conductivity of the soil is calculated as weighted power mean from the conductivities and volumetric fractions of the soil constituents water, ice, air, mineral and organic."*

*Following Westermann et al. (2013, 2016) the according equation describes the soil thermal conductivity (k)*

$$k = \left( \sum_{\alpha} \Theta_{\alpha} \sqrt{k_{\alpha}} \right)^2$$

*with the volumetric fractions (θα) of water, ice, air, mineral and organic. This parameterization has been used as standard in CryoGrid for a number of publications (i.e. Nitzbon et al. 2019, 2020). The temperature-dependence of the thermal conductivity, which gives rise to the thermal offset between ground surface and permafrost temperatures (Osterkamp and Romanovsky, 1999), is contained in the temperature-dependent water and ice contents as detailed in Sect. 2.2 in Westermann et al. (2013). This parameterization above is chosen for simplicity (e.g., de Vries, 1952; Farouki, 1981 describe other parameterizations), as reliable recommendations for a particular conductivity model are lacking for permafrost areas and as our chosen parameterization allows us to successfully reproduce observed annual freeze and thaw cycles at permafrost sites under differing environmental conditions (Westermann et al., 2013).*

Figure 3: What are the error bars showing? What is the input to calculate error bars (hourly, daily, weekly data)? It seems as if the variability (i.e. error bars) for the observed grassland turbulent energy fluxes is much larger than the modelled variability.

*We agree with the reviewer that this has not been made clear. Whiskers in modeled data show the standard deviation based on daily averaged data. Further, the whiskers in the measured data at the grassland site were based on half-hourly data. We thank the reviewer for pointing this out and* **have adapted this for the measured data at the grassland site to be based on daily values as well.** *This provides more comparable variability values. We modify the figure caption accordingly to explain what the whiskers show:* **"Surface energy balance for snow-covered (28.10.2018-27.04.2019) and snow-free (10.10.2019-27.10.2019 and 28.04.2019- 10.10.2019) periods at the ground surface of grassland and forest and at the top of the canopy of forest (Forest TOC). Shown are the net radiation (Qnet), sensible (Qh), latent (Qe) and storage heat flux (Qs) for the model runs of the forest and grassland site as well as the measured values at the grassland site. The bars indicate mean values while the whiskers show the corresponding standard deviations."**

Table 2: Is this the same data as shown in Figure 5? If so, the authors could think about showing only one of them (figure or table).

*We agree with the reviewer that this is indeed a repetition; therefore,* **the values shown in Table 2 are added to Figure 5 directly. Table 2 has been removed.**

Figure 6: Most of the model-observation comparison is of qualitative nature. The authors could add some performance metrics (e.g., RMSE, R2, bias: : :)

*We thank the reviewer for this suggestion and agree that performance metrics would be helpful to evaluate the model performance. **We will add performance metrics such as $R^2$ values to describe the significance of the differences between measurements and model outcomes as well as between the forest and grassland sites to figure 4, 5 and 6.***

Page 18, line 12: The authors report a bias in modelled GST during the winter. Since GST measurements are available, the forest-grassland comparison could be more meaningful if it were based on observational data.

*These numbers are provided in Figure 5 (and formerly Table 2). Based on this suggestion the measured differences will now also be provided in the text (p. 18, l. 10): **"Our results are in agreement with these observations, but further demonstrate that the impact of mixed boreal forest on the GST is strongest during the snow period and the summer peak with the warmest months. Our model reveals an average of 6.5°C higher GST during the snow-covered period and 1.5°C lower GST during the snow-free period. Measurements reveal an average of 2°C higher GST during the snow-covered period and 2.3°C lower GST during snow-free periods."***

Page 22, line 17: Could lateral flow of water contribute to differences in ground water content?

*We would like to thank the reviewer for this important question. Indeed lateral water flow could contribute to the differences in actual measured ground water content. This process is neglected in this baseline, one-dimensional model set-up where we try to investigate the influence of forest on the surface energy balance of the ground. Recently, Nitzbon et al. (2019) have integrated lateral fluxes of heat, water and snow in the CryoGrid scheme. Higher forested ground water content was measured in point measurements taken in 2018, but no year-long ground water content measurements are available for our forest site, therefore this is not discussed in detail within this study. We add this information to p.10, l.12: **"Lateral water fluxes are neglected in this baseline, one-dimensional model set-up."** We further add the following statement to the discussion (p. 22, l.7): **"Further, one aspect not represented in the model is the moisture transport and migration in frozen ground or the forming of ice lenses. Lateral water flow and snow redistribution may be important processes to be investigated in the future since they can strongly modify the thermal regime."***

**References**

Chen, X., L. Vierling , D. Deering & A. Conley (2005) Monitoring boreal forest leaf area index across a Siberian burn chronosequence: a MODIS validation study, International Journal of Remote Sensing, 26:24, 5433-5451, https://doi.org/10.1080/01431160500285142

Cosenza, P., Guerin, R., and Tabbagh, A. (2003): Relationship between thermal conductivity and water content of soils using numerical modelling, Eur. J. Soil Sci., 54, 581–588

de Vries, D. (1952): The thermal conductivity of soil, Mededelingen van de Landbouwhogeschool te Wageningen, 52, 1–73

Estop-Aragonés, C., Cooper, M. D. A., Fisher, J. P., Thierry, A., Garnett, M. H., Charman, D. J., et al. (2018). Limited release of previously-frozen C and increased new peat formation after thaw in permafrost peatlands. Soil Biology and Biochemistry, 118(December 2017), 115–129. https://doi.org/10.1016/j.soilbio.2017.12.010

Farouki, O. (1981): The thermal properties of soils in cold regions, Cold Reg. Sci. Technol., 5, 67–75

Nitzbon, J., Langer, M., Westermann, S., Martin, L., Aas, K. S., and Boike, J. (2019): Pathways of ice-wedge degradation in polygonal tundra under different hydrological conditions, Cryosphere, 13, 1089–1123, https://doi.org/10.5194/tc-13-1089-2019

Nitzbon, J., Westermann, S., Langer, M., Martin, L. C. P., Strauss, J., Laboor, S., and Boike, J. (2020): Fast response of cold ice-rich permafrost in northeast Siberia to a warming climate, Nature Communications, 11, https://doi.org/10.1038/s41467-020-15725-8

Ohta, T., Hiyama, T., Tanaka, H., Kuwada, T., Maximov, T. C., Ohata, T., and Fukushima, Y.: Seasonal variation in the energy and water exchanges above and below a larch forest in eastern Siberia, Hydrological Processes, 15, 1459–1476, https://doi.org/10.1002/hyp.219,2001

Osterkamp, T. and Romanovsky, V. (1999): Evidence for warming and thawing of discontinuous permafrost in Alaska, Permafrost Periglac., 10, 17–37

Westermann, S., Schuler, T., Gisnås, K., and Etzelmüller, B. (2013): Transient thermal modeling of permafrost conditions in Southern Norway, The Cryosphere, 7, 719–739, https://doi.org/10.5194/tc-7-719-2013

---

## Author Comment (AC2) · 19 Sep 2020

Reviewer comments

*Author response,* **Changes made in manuscript**

**------------------------**

**Anonymous Referee #2**

Comments on "Variability of the surface energy balance in permafrost underlain boreal forest" submitted by Simone Maria Stuenzi et al. to Biogeosciences

In this manuscript, the authors developed a biophysical atmosphere-land interaction model (CryoGrid) and examined its performance by comparing with observations at a mixed forest in East Siberia. By coupling with a multi-layer canopy scheme (CLM-ml v0), they accounted for the effects of vegetation canopy on surface processes. They conducted a series of simulations with different land covers (forest and grassland) and different leaf area index or canopy density. They found that coverage with high leaf area index affects surface energy budget, such as solar radiation transfer and insulation, and that it could play important roles in snow and permafrost dynamics such as active layer thickness. I agree that terrestrial ecosystems in northern high latitude is remarkable in terms of climate change, especially permafrost thawing that is thought as one of the tipping elements of the Earth system. Then, developing advanced models simulating boreal ecosystem processes is highly important.

*We thank the reviewer for the positive perspective on the general topic of our study. We absolutely share the opinion that it is extremely relevant to study boreal forest and permafrost interactions under amplified climate change. Further, we gratefully acknowledge all critical comments, which help us to improve our manuscript. We have thoroughly gone through all comments and suggestions made by the reviewer. Please note that any changes and additions we propose for the revised manuscript are highlighted in* **bold.**

Nevertheless, I could not find out what is the original contribution of this study. For example, the model developed in this study seems similar to previous models such as CLM, ORHIDEE, LPJ etc. These previous models have already implemented leaf phenology and dynamic vegetation; at least some versions include permafrost dynamics.

*We thank the reviewer for this critical remark on our manuscript and take this as an opportunity to clarify the scientific questions that we aim to answer with the performed modelling exercise. We appreciate your concern that previous models have already implemented leaf phenology or dynamic vegetation and would like to elaborate on this accordingly. The mentioned models such as Orchidee-Can (Chen et al. 2016), Lund-Potsdam-Jena (LPJ DGVM) (Beer et al. 2007), CLM (Levis et al., 2004) or NEST (Zhang et al. 2003) and SiBCliM (Tchebakova et al. 2009) are discussed in the Introduction section on page 2 starting on line 22. Certainly, these models include some sort of dynamic or static vegetation parameterization, leaf phenology and some models also permafrost dynamics. The focus thereof lies on the forest establishment and mortality (Sato et al. 2016), unfrozen vs. frozen ground and fire disturbances (Zhang et al. 2011) or the evolution of the vegetation*

carbon density under diverse warming scenarios (Beer et al. 2007). The multilayer canopy module presented here has been developed for a future integration in the CLM scheme, but has not been used in permafrost underlain boreal forests. Further, the presented model is an advancement, as it is specifically developed to study heat transfer processes with both, high-resolution permafrost ground and a high-resolution vegetation canopy. This allows us to quantify the interactions between detailed canopy structures and permafrost. To amplify the importance and uniqueness of our study we propose to add the following section to the introduction (p.2, l. 31):

**"While all of these studies have significantly improved our understanding of essential mechanisms in boreal permafrost ecosystems, it is important to further understand how a forest canopy affects the thermal state and the snow regime of the ground, especially amid ongoing shifts in forest composition (Loranty et al., 2018). The existing model set-ups are often static or not able to capture important processes such as the vertical canopy structure or the leaf physiological properties, which determine the energy transfer between the top of the canopy atmosphere and the ground. These general canopy models focus on reproducing the forest properties, but they have not been evaluated much for permafrost settings and with respect to the impact of forest on permafrost. To our knowledge, so far, none of the existing models is able to capture the important processes of the vertical canopy structure in combination with a physically-based, highly advanced permafrost model. The novel model introduces a robust radiative transfer scheme through the canopy for a detailed analysis of the vegetation's impact on the hydro-thermal regime of the permafrost ground below. This allows us to quantify the surface energy balance dynamics below a complex forest canopy and its direct impact on the hydro-thermal regime of the permafrost ground below."**

Further, we have made the conclusions and this study's original contribution clearer by adding the following key points (p.23, l.1):

**"This study presents a specific application of a coupled multilayer forest-permafrost model to investigate the energy transfer and surface energy balance in permafrost underlain boreal forest of Eastern Siberia. The comparison of measured and modeled GST at a mixed forest and a grassland site, the comparison of the modeled and measured radiation fluxes at the grassland site, as well as the comparison of modeled and measured radiation fluxes at an external study site, justify the use of the physically-based modeling approach to investigate the thermal regime and surface energy balance in this complex ecosystem. Based on this modeling exercise and field measurements, we investigate the thermal conditions of two landscape entities as they typically occur in the boreal zone. In regard to the forests insulation effect on permafrost and ongoing land cover transition this study delivers important insights into the range of spatial differences and possible temporal changes that can be expected following landscape changes such as deforestation through fires, anthropogenic influences and afforestation in currently unforested grasslands or densification of forested areas. The detailed vegetation model successfully calculates the canopy radiation and water budgets, leaf fluxes, as well as canopy**

*turbulence and aerodynamic conductance. These canopy fluxes alter
the below-canopy surface energy balance, the ground thermal
conditions and the snow cover dynamics. We find a strong dampening
effect of 19°C on the annual ground surface temperature amplitude of
the permafrost."*

Another concern on this study is that the authors used a very limited amount of observations.
Especially, they validated the model performance to capture temporal variability only for
surface temperature. Because many flux measurement sites are operating and providing a
variety of observational data, I highly recommend validating the model by using a larger
number of biophysical variables including energy and water fluxes.

*We appreciate the suggestion on using a larger number of biophysical
variables including energy and water fluxes. In accordance with the
comments of reviewer 1, we agree that our study requires a more
extensive validation based on surface energy balance measurements.
Thus, we extended the model validation to an additional site for
which extensive surface energy balance measurements are available in
order to demonstrate the capabilities of our model. Not many such
flux measurement sites are available in Eastern Siberia, to our
knowledge; no such site exists around our study side (Nyurba/Vilnuy
area). We suggest using existing and available data from the rather
near, well-documented and well-studied research site Spasskaya-Pad
at 62°14'N, 129°37'E. We have been provided meteorological and
radiation data from beneath and above the larch-dominated forest
canopy for 2018 through the Arctic Data Archive system (ADS). This
data is used for additional model validation. To justify our model
we suggest adding this additional validation site to the appendix of
our manuscript.*

*As described in our answer to reviewer 1 we have set-up and ran a 5-
year simulation for this site, using ERA-interim forcing data for
the coordinate above and a summer LAI of 3.66 $m^2m^{-2}$ (following the
measurement-based LAI in Ohta et al. 2001) and a tree height of 18
m. Since the study site is larch-dominated we have now implemented a
simple leaf-off parameterisation which is used here. This results in
a winter LAI of 1.66 $m^2m^{-2}$ (again based on Ohta et al. 2001) and a
leaf-off period from 10. October - 10. April.*

*Preliminary analyses of simulation results show a good fit with the
modeled surface energy balance. The following preliminary paragraph
is added to the Appendix, in a novel section "External validation
site "Spasskaya Pad"":* **"For further validation of the model
performance we use existing and available data from the rather near,
well-documented and well-studied research site Spasskaya-Pad at
62°14'N, 129°37'E. Through the Arctic Data Archive system (ADS) we
have been provided meteorological and radiation data from beneath
and above the larch-dominated forest canopy for 2018. Therefore, we
have set-up and ran a 5-year simulation for this study site, using
ERA-interim forcing data for the coordinate above and a summer LAI
of 3.66 $m^2m^{-2}$ (following the measurement-based LAI in Ohta et al.
2001) and a tree height of 18 m. Since the study site is larch-
dominated we have now implemented a simple leaf-off parameterisation
which is used here. This module allows for a leaf-off period from
fall to spring. This results in a winter LAI of 1.66 $m^2m^{-2}$ (again
based on Ohta et al. 2001) and a leaf-off period from 10. October -
10. April. Preliminary analyses of simulation results show a good**

*fit with the modeled surface energy balance and justify the use of
the model in the current version."*

Also, I have a concern about the overly simplified simulation conditions, such as the lack of seasonal change in leaf area index in forest. Though in situ data were not available, at present, we can obtain time-series of leaf area index from satellite remote sensing as a good proxy.

*We thank the reviewer for this critical remark concerning the canopy
phenology. As discussed in the "Applicability and model limitations"
section (p.21, l.14), the development and implementation of a
phenology module was considered to be out of scope for this study,
mainly because of the little deciduous taxa (only 7%) at the chosen
study site. As explained in our response to the previous comment, **a
phenology module allowing for a leaf-off period from fall to spring
has now been implemented for the newly included study site at the
Spasskaya Pad research station.** Here, deciduous larch is dominant
and a phenology module is therefore highly important.*

Based on the inadequacy of scientific insights and model simulations, I cannot recommend this manuscript to accept for publication in the present form. The manuscript is too descriptive and needs more model validation with observations. Also, the authors need to devise model simulations to make insightful discussions.

We thank the reviewer for her/his opinion on our submitted
manuscript and would like to summarize the substantial changes and
adaptations described in detail above and in the responses to
Reviewer 1. The main changes we propose to improve our manuscript
are:

- Adding a further validation site (Spasskaya Pad) with an
  extensive record of energy flux measurements which is, to our
  knowledge, a unique data record for a forested study site in
  Eastern Siberia. Due to a higher component of deciduous taxa,
  we have implemented a canopy phenology module to simulate the
  specific traits of deciduous taxa. With this additional
  validation site we justify the use of our current model set-
  up.
- We would also like to point out changes in the introduction and
  conclusion sections (see above) which now clarify the novelty
  of our modeling exercise and the importance of the found and
  described insights in severely understudied high-latitude,
  permafrost-underlain boreal forest areas.

Minor points
Page 3 Section 2.1: Can you give information on vegetation conditions, such as leaf area index and tree density?

*Following our response to reviewer 1, we recognize that the
information given on LAI estimation on p.10, l.26 is insufficient,
therefore we have modified the paragraph to the following, more
detailed description: **"LAI can be estimated from satellite data,
calculated from below-canopy light measurements or by harvesting
leaves and relating their mass to the the canopy diameter. Ohta et
al. (2001) have described the monitored deciduous-needleleaf forest***

*site at Spasskaya Pad research station, which has comparable climate conditions but is larch-dominated. The value of the tree plant area index (PAI), obtained from fish-eye imagery and confirmed by litter fall observations, varied between 3.71 $m^2m^{-2}$ in the foliated season and 1.71 $m^2m^{-2}$ in the leafless season. This value does not include the ground vegetation cover. Further, Chen et al. (2005) compared ground-based LAI measurements to MODIS values at an evergreen-dominated study area (57.3° N, 91.6° E) south-west of the region discussed here, around the city of Krasnoyarsk. The mixed forest consists of spruce, fir, pine and some occasional hardwood species (birch and aspen). They find LAI values between 2 $m^2m^{-2}$ and 7 $m^2m^{-2}$. To assess the LAI we use data from literature and the experience from the repeated fieldwork at the described site. Following Kobayashi et al. (2010) who conducted an extensive study using satellite data, the average LAI for our forest type is set to 4 $m^2m^{-2}$ and stem area index (SAI) is set to 0.05 $m^2m^{-2}$, resulting in a plant area index (PAI)of 4.05 $m^2m^{-2}$ and 9 vegetation layers for model simulations."*

*To further answer this question, leaf area index has not been measured explicitly, but is described in the study by Kobayashi et al. (2010) and can also be estimated from satellite imagery such as the Copernicus LAI 300m Version 1 product (available through https://land.copernicus.vgt.vito.be/PDF/portal/Application.html#Home ). Accordingly, LAI at our study site is between 3 and 4 $m^2m^{-2}$. Tree density could be estimated from the vegetation survey mentioned on page 5, line 20.*

Page 7 Line 3: I am not sure how within-canopy wind profile was parameterized and simulated by the multi-layer canopy model. Can you explain briefly, because this is an important feature of multi-layer canopy models?

*This is not discussed in particular here, but described in detail in Bonan et al. 2018. We add this information to our manuscript (p.6,l.21):* **"The within-canopy wind profile is calculated using above- and within-canopy coupling with a roughness sublayer (RSL) parameterization (see Bonan et al. 2018 for further detail)."**

References

Beer, C., Lucht, W., Gerten, D., Thonicke, K., and Schmullius, C. (2007): Effects of soil freezing and thawing on vegetation carbon density in Sierian burberia: A modeling analysis with the Lund-Potsdam-Jena Dynamic Global Vegetation Model (LPJ-DGVM), Global Biochemical Cycles, 21, https://doi.org/10.1029/2006GB002760

Bonan, G. B., Patton, E. G., Harman, I. N., Oleson, K. W., Finnigan, J. J., Lu, Y., and Burakowski, E. A. (2018): Modeling canopy-induced turbulence in the Earth system: A unified parameterization of turbulent exchange within plant canopies and the roughness sublayer (CLM-ml v0), Geoscientific Model Development, 11, 1467–1496, https://doi.org/10.5194/gmd-11-1467-2018

Chen, X., L. Vierling , D. Deering & A. Conley (2005) Monitoring boreal forest leaf area index across a Sibn chronosequence: a MODIS validation study, International Journal of Remote Sensing, 26:24, 5433-5451, https://doi.org/10.1080/01431160500285142

Chen, Y., Ryder, J., Bastrikov, V., McGrath, M. J., Naudts, K., Otto, J., Ottlé, C., Peylin, P., Polcher, J., Valade, A., Black, A., Elbers, J. A., Moors, E., Foken, T., van Gorsel, E., Haverd, V., Heinesch, B., Tiedemann, F., Knohl, A., Launiainen, S., Loustau, D., Ogée, J., Vessala, T., and Luyssaert, S. (2016): Evaluating the performance of land surface model ORCHIDEE-CAN v1.0 on water and energy flux estimation with a single- and multi-layer energy budget scheme, Geoscientific Model Development, 9, 2951–2972, https://doi.org/10.5194/gmd-9-2951-2016

Kobayashi, H., Delbart, N., Suzuki, R., and Kushida, K.: A satellite-based method for monitoring seasonality in the overstory leaf area index of Siberian larch forest, Journal of Geophysical Research: Biogeosciences, 115, 1–14, https://doi.org/10.1029/2009JG000939

Levis, S., Bonan, G. B., Vertenstein, M., & Oleson, K. W. (2004). The Community Land Model's Dynamic Global Vegetation Model (CLM-DGVM): Technical Description and User's Guide. NCAR/Tn-459+Ia. https://doi.org/10.5065/D6P26W36

Loranty, M. M., Abbott, B. W., Blok, D., Douglas, T. A., Epstein, H. E., Forbes, B. C., Jones, B. M., Kholodov, A. L., Kropp, H., Malhotra, A., Mamet, S. D., Myers-Smith, I. H., Natali, S. M., O'donnell, J. A., Phoenix, G. K., Rocha, A. V., Sonnentag, O., Tape, K. D., and Walker, D. A. (2018): Reviews and syntheses: Changing ecosystem influences on soil thermal regimes in northern high-latitude permafrost regions, Biogeosciences, 15, 5287–5313, https://doi.org/10.5194/bg-15-5287-2018

Ohta, T., Hiyama, T., Tanaka, H., Kuwada, T., Maximov, T. C., Ohata, T., and Fukushima, Y.: Seasonal variation in the energy and water exchanges above and below a larch forest in eastern Siberia, Hydrological Processes, 15, 1459–1476, https://doi.org/10.1002/hyp.219,2001

Sato, H., Kobayashi, H., Iwahana, G., and Ohta, T. (2016): Endurance of larch forest ecosystems in eastern Siberia under warming trends, Ecology and Evolution, 6, 5690–5704, https://doi.org/10.1002/ece3.2285

Tchebakova, N. M., Parfenova, E., and Soja, A. J. (2009): The effects of climate, permafrost and fire on vegetation change in Siberia in a changing climate, Environmental Research Letters, 4, https://doi.org/10.1088/1748-9326/4/4/045013

Westermann, S., Schuler, T., Gisnås, K., and Etzelmüller, B. (2013): Transient thermal modeling of permafrost conditions in Southern Norway, The Cryosphere, 7, 719–739, https://doi.org/10.5194/tc-7-719-2013

Zhang, Y., Chen, W., and Cihlar, J. (2003): A process-based model for quantifying the impact of climate change on permafrost thermal regimes, Journal of Geophysical Research D: Atmospheres, 108, https://doi.org/10.1029/2002JD003354

Zhang, N., Yasunari, T., and Ohta, T. (2011): Dynamics of the larch taiga-permafrost coupled system in Siberia under climate change, Environmental Research Letters, 6, https://doi.org/10.1088/1748-9326/6/2/024003

---

## Author Response (AR1)

Stuenzi et al. use measurements and modelling of surface energy balance processes for a boreal forest and grassland site in Siberia to explore the impact of vegetation characteristics on ground thermal dynamics and snow cover. They couple a 1-D land surface model to a multilayer canopy model to account for radiative transfer through the canopy and for soil thaw dynamics. They find that the forest canopy efficiently reduces solar radiation at the forest floor causing slower soil warming and thaw and delayed snow melt.

The manuscript aims to better describe how vegetation interacts with soil thermal regimes in the continuous permafrost zone of Siberia. The authors provide an important and detailed description of the most relevant vegetation-ground interactions, which can help to better understand permafrost responses in a warming climate. One concern is that no radiation, snow, or Bowen ratio measurements were available at the forest site. I know of the difficulties of setting up long-term observations in such environments, but the performance of the model simulations for the forest site - which is the major focus of this paper – cannot be properly assessed without these observations.

The authors discuss this shortcoming in the discussion, but – in my opinion – they need to justify better then why the modelling results should be trusted. The modelled forest GST seem to reasonably fit the measured GST, but modelled soil thaw is delayed compared to observations. This raises the question if the modelling results regarding snow phenology are meaningful.

The manuscript provides a good overview and description of the relevant surface energy balance processes, but the authors could highlight how their study advances our understanding of surface energy balance processes in the forested permafrost zone. Bonan and Shugart (1989) outlined many of the relevant interactions and, for example, Chasmer et al. (2011, DOI:

10.1002/ppp.724) present results on forest canopy effects on radiative processes in a permafrost environment.

*We thank the reviewer for this positive overall evaluation of our study and for taking the time to review our manuscript. We have worked through all of the posed questions and suggestions made by the reviewer, which has improved our manuscript. Please note that any changes and additions to the text that we propose for the revised manuscript are highlighted in* **bold.**

*With this manuscript we, indeed, aim at improving our understanding of how the vegetation interacts with the solid thermal regime in the continuous permafrost zone of eastern Siberia. We would like to thank the reviewer for acknowledging the importance of this detailed description of some of the most relevant vegetation-ground interactions in understanding permafrost responses to a warming climate.*

*Indeed, setting up such long-term measurements in these environments is highly challenging, but we agree that the model performance cannot be assessed completely without further integration of measurements. The model does successfully reproduce the measured ground surface temperatures (GST) of the monitored year (August 2018 – August 2019). Nevertheless, GST does not provide a full picture of the surface energy balance. Therefore, we suggest to further validate the model performance with additional measurements. Acquisition of sub-canopy radiation data is, as recognized by the reviewer, highly challenging and was unavailable within the scope of the field work underlying the presented study. Since not many monitoring sites exist around Siberia, we suggest using existing and available data from the rather near, well-documented and well-studied research site Spasskaya-Pad at 62°14'N, 129°37'E. Through the Arctic Data Archive system (ADS) we have been provided meteorological and radiation data from beneath and above the larch-dominated forest canopy for 2018. This data is used for additional model validation. Unfortunately, snow-depth measurements are not readily available. We have added this additional model validation to the appendix of our manuscript since it is a rather technical aspect which is not directly related to our major study site. A detailed analysis of this validation run is presented in the following and in the revised manuscript. Detailed analyses show a good fit with the novel measurement data and therefore justify the use of the model in the current version.* ***We add the following paragraph to the section "Model validation and in-situ measurements" (p.13, l.270):***

*"We acknowledge that the target variable GST does not provide a full picture of the surface energy balance. Therefore, we further validate the model performance with additional measurements from an external study site. Through the Arctic Data Archive system (ADS) we have been provided meteorological and radiation data from beneath and above the larch-dominated forest canopy in Spasskaya-Pad for 2017-2018 (Maximov et al., 2019). This data is used for additional model validation and is added to the appendix of our manuscript. Overall our analysis reveals a satisfactory agreement between modeled and measured components of the surface energy balance below the canopy. Thus, we argue that the performance of the model at the external study site justifies its application at the primary study site in Nyurba where below canopy fluxes were not acquired (see Appendix C)."*

*Respectively, we add the following section to Appendix C (p.27): "*

*Further validation of the model performance is performed for a well-studied research site in Spasskaya-Pad at N 62.14°, E 129.37°. This additional validation site is located 581 km from our primary study site, but allows to validate further model variables due to additional observational data. Spasskaya-Pad is a continuous permafrost region and the active layer depth is about 1.2 m in larch-dominated forests. Main tree species is Dahurian larch (Larix gmelinii) with a stand density of 840 trees/ha. Understory vegetation (Vaccinium) is dense and 0.05 m high. In 1996 a 32m observation tower was installed (Ohta et al., 2001) in larch dominated forest. Through the Arctic Data Archive system (ADS, https://ads.nipr.ac.jp/) we have been provided with the most recent, available meteorological and radiation data from beneath and above the larch-dominated forest canopy for the time-period 2017-2018 (Maximov et al., 2019). Each variable used here is measured at exactly 5-min intervals, except radiation. Ventilated shelters cover air temperature and humidity sensors. Net all-wave radiation and four components of radiation are measured every minute, and the data loggers record average,maximum and minimum values. Upward and downward long-wave radiation is corrected using the sensed temperature at domes and sensor bodies. Ground temperature is measured at seven depths, and soil moisture at five depths. A more detailed description of the sensors can be found in Table 1 in Ohta et al. (2001). We have set-up and ran a 6-year simulation for this study site (2013-2019), using ERA-interim forcing data using the grid cell closest to the coordinate N 62.14°, E 129.37°. The measurement tower is situated in larch-dominated forest and we have now implemented a simple leaf-off parameterisation which*

*is used here. Based on Ohta et al. (2001) we define a partial leaf-off period from 10. October - 30. May, resulting in a winter LAI of 0.5 $m^2m^{-2}$. Summer LAI is 1.9 $m^2m^{-2}$ and a tree height of 18 m. For a comprehensive model validation we use the same soil parameters for this external study site. Table A3 summarizes all parameter choices for soil stratigraphies and Table A4 summarizes constants used. The subsurface stratigraphy extends to 100 m below the surface where the geothermal heat flux is set to 0.05 $W/m^{-2}$ (Langer et al., 2011b). The ground is divided into separate layers in the model. The uppermost 8 m have a layer thickness of 0.05 m, followed by 0.1 m for the next 20 m, 0.5 m up to 50 m and 1 m there after. The remaining CryoGrid parameters were adopted from previous studies using CryoGrid (Table A2) (Langer et al., 2011a, b, 2016; Westermann et al., 2016; Nitzbon et al., 2019, 2020). Again, we use ground surface temperature (GST) as one of the target variables for model validation, measured at 0.2 m. In addition we use air temperature below the canopy, measured at the height of 1.2 m, net radiation ($Q_{ne}t$), latent ($Q_e$) and sensible ($Q_h$) heat flux, and incoming ($S_{in}$) and outgoing ($S_{out}$) shortwave radiation flux at the ground surface as target variables allowing a comprehensive validation of the modeled heat and moisture exchange processes within and below the canopy.*

*We assess the surface energy balance by comparing the median weekly values of modeled and measured net radiation (Qnet), sensible heat flux ($Q_h$), latent heat flux ($Q_e$), and incoming ($SW_{in}$) and outgoing ($SW_{out}$) solar radiation at the forested site (Fig. C1).*

*Modeled turbulent fluxes at forest ground are small for snow-covered periods and measured data are not available during this period. Modeled and measured sensible heat flux in the snow-free period differ by 0.1 $Wm^{-2}$ only. Modeled latent heat flux is only a fourth of the measured value and therefore underestimated in our model. Modeled net radiation in the snow-free period (25.7 $Wm^{-2}$) is slightly above measured net radiation (19.4 $Wm^{-2}$). For the snow-covered period modeled net radiation is slightly below the measured median value. The incoming shortwave radiation measured and modeled for the forest site fit well with differences well below 10 $Wm^{-2}$. The standard deviation of measured values is higher for all variables except snow-covered net radiation.*

*In a second step, we compare the modeled and measured annual, snow-free and snow-covered median GST and air temperature below the canopy to understand the overall model performance regarding the*

*thermal regime of the surface and the ground and the relative temperature differences between the model and measurements (Fig. C2). The highest deviation between modeled and measured temperatures is found in the GST of the snow-free period. Here, the model shows a cold bias of 2°C. For the snow-covered period the difference is 1.8°C. For the air temperature below the canopy the difference between modeled and measured in the snow-free period is 1.5°C, for snow-covered the difference is again 1.8°C. This falls into the range of 1.5-2°C that are commonly used for validation purposes (Langer et al., 2013, Westermann et al., 2016).*

*Overall our analysis reveals a satisfactory agreement between modeled and measured components of the surface energy balance below the canopy. Thus, we argue that the performance of the model at the external study site justifies its application at the primary study site in Nyurba where below canopy fluxes were not acquired."*

*Fig. C1 and Fig. C2 are added with the following caption for Fig. C1: "Modeled (grey) incoming and outgoing solar radiation ($S_{in}$, $S_{out}$) and turbulent fluxes ($Q_h$, $Q_e$, $Q_{net}$) for snow-covered (28.10.2017-27.04.2018, above) and snow-free (10.10.2017-27.10.2017 and 28.04.2018-10.10.2018, below) periods at the ground surface of forest." and for Fig. C2: "Left, modeled (grey) and measured (black) air temperature (°C) below the canopy. Right, ground surface temperature (°C). Both for snow-covered (28.10.2017-27.04.2018, above) and snow-free (10.10.2017-27.10.2017 and 28.04.2018-10.10.2018, below) periods at the forest site in Spasskaya Pad."*

*Based on the measured radiation data from Spasskaya Pad we have redone the original simulation, now including the ERA-Interim forcing variable cloud cover to be able to differentiate between direct and diffuse radiation components. **We add the following paragraph to the Appendix B (p.26): "Cloud cover (N) allows us to use a simple approach to differentiate the incoming shortwave radiation ($S_{in}$) into diffuse***

$$S_{in, diffuse} = S_{in} * \left( 0.3 + 0.7 * \frac{N^2}{8} \right) ,$$

**and direct**

$$S_{in, direct} = S_{in} - S_{in, diffuse}$$

**components, based on Younes et al., 2007."**

*We have updated the figures, numbers and statistics in the main text accordingly. Additionally, we add the following paragraph to the main text (p.12, l.246):* **"The meteorological forcing data required by the model include: air temperature, relative humidity, air pressure, wind speed, liquid and solid precipitation, incoming short- and longwave radiation and cloud cover. ERA-Interim data for the coordinate N 63.18946°, E 118.19596° were used to obtain forcing data for the total available period from 1979 to 2019 (Simmons et al., 2007)."**

*We further thank the reviewer for the suggested literature such as Bonan and Shugart (1989) who outlined relevant interactions and Chasmer et al. (2011) that present results on forest canopy effects on radiative processes in a permafrost environment. Bonan et al. (1989) is one of our main sources providing the overall framework of this study and is discussed on p.2, l.32, p.18, l.342 and on p.19, l.381 and l.384.*

*We have now carefully studied the article by Chasmer et al. (2011) which does present very interesting results for interaction processes in a discontinuous permafrost zone in Canada. From this study we have learned that vegetation on the edges of permafrost plateaus tends towards reduced fractional canopy cover (by up to 50%) and reduced canopy heights (by 16–30%). The reduced biomass can cause a positive feedback because of lower canopy shading (up to 1h per day less), which leads to an increase in incident radiation at the ground (+16% at open sites) and higher longwave radiation losses (+74% at open plateau sites). We will incorporate this important reference in the following sentence in the introduction (p.2, l.32):* **"Changing climatic conditions can promote an increasing active layer depth or trigger the partial disappearance of the near surface permafrost. Further, extensive ecosystem shifts such as a change in composition, density or the distribution of vegetation (Holtmeier and Broll, 2005; Pearson et al., 2013; Gauthier et al., 2015; Kruse et al., 2016; Ju and Masek, 2016) and resulting changes to the below- and within-canopy radiation fluxes (Chasmer et al., 2011) have already been reported."**

Other comments

Page 2, Line 19: In some cases, permafrost thaw and forest loss can also lead to increased

$CO_2$ uptake as shown for thawing ice-rich permafrost in northwestern Canada and Alaska.

*We agree with the importance of this finding and suggest to rearrange the sentence to the following (p.2, l.35): "Changes to the vegetation – permafrost dynamics can have a potentially high impact on the numerous feedback mechanisms between the two ecosystem components.* **Increased soil carbon release from thawing permafrost through the delivery of soil organic matter to the active carbon cycle (Schneider Von Deimling et al., 2012) is modified by vegetation changes, which can compensate for carbon losses due to an increased $CO_2$ uptake (as observed at ice-rich permafrost sites in northwestern Canada and Alaska, Estop-Aragonés et al., 2018) or even further accelerate total carbon loss (Romanovsky et al., 2017)"**

Page 3, Line 25: By how much has the summer precipitation decreased?

*We agree that this is an important question and we propose to add the following information (p.3, l.82):* **"Annual precipitation showed an increasing trend from 1900 until 1990, mainly due to an increase in wintertime precipitation. Between 1995 and 2002, summertime precipitation has decreased by -16.9 mm in August and -4.2 mm in July (see table 1 in Hayasaka (2011) for further details)."**

Page 5, Line 20: How was tree height estimated? Are there any ground-based LAI measurements in similar forest types?

*We thank the reviewer for this question and have added the following information to the manuscript (p.6, l.116):* **"In a vegetation survey along a 150 m transect from the grassland into the forest, the tree height of every tree within a 2 m distance was estimated. Trees <2 m were measured with a measuring tape, trees >2 m were measured with a clinometer or visually estimated after repeated comparisons with clinometer measurements."**

*Further, we recognize that the information given on LAI estimation on p.12, l.230 is insufficient, therefore we have modified the paragraph to the following, more detailed description:* **"LAI can be estimated from satellite data, calculated from below-canopy light measurements or by harvesting leaves and relating their mass to the the canopy diameter. Ohta et al. (2001) have described the monitored deciduous-needleleaf forest site at Spasskaya Pad research station, which has comparable climate conditions but is larch-dominated. The value of the tree plant area index (PAI), obtained from fish-eye imagery and confirmed by litter fall observations, varied between 3.71 in the foliated season and 1.71 in the leafless season. This value does**

*not include the ground vegetation cover. Further, Chen et al. (2005) compared ground-based LAI measurements to MODIS values at an evergreen-dominated study area (57.3° N, 91.6° E) south-west of the region discussed here, around the city of Krasnoyarsk. The mixed forest consists of spruce, fir, pine and some occasional hardwood species (birch and aspen). They find LAI values between 2 and 7. To assess the LAI we use data from literature and the experience from the repeated field work at the described site. Following Kobayashi et al. (2010) who conducted an extensive study using satellite data, the average LAI for our forest type is set to 4 $m^2 m^{-2}$ and stem area index (SAI) is set to 0.05 $m^2 m^{-2}$, resulting in a plant area index (PAI) of 4.05 $m^2 m^{-2}$ and 9 vegetation layers for model simulations."*

Table 1: Perhaps, the equations in Table 1 could be shown in the table itself or at least qualitatively described. As it is now, the content of the table is not easy to grasp.

*We agree with the reviewer that table 1 has little additional value at the moment.* **We have added qualitative descriptions of Eq. 2 and 4 to the table directly.**

Page 6, Line 4: Why did the authors choose the PFT "deciduous needleleaf forest"? They mention that the site is dominated by Picea obovata (92%). Wouldn't an evergreen needleleaf parameterisation be more adequate? Also, if using the PFT "deciduous needleleaf forest", wouldn't it be necessary to include a phenology module? This is partly discussed later in the manuscript, but should be already mentioned here.

*We thank the reviewer for this remark. The PFT evergreen needleleaf (NET) is used for the simulations.* **This has been corrected in table A5 and on page 6.** *We later discuss the implementation of the needle-tossing (p.23, l.449):* **"To simulate the needle-tossing of deciduous larch, we have incorporated a leaf area index threshold between needle-tossing and leaf-out (10. October - 30. May) for simulations at our external validation site at Spasskaya-Pad (see Appendix C). This tunes the model towards a more detailed representation of larch-dominated forests, which are particular to the secondary study site and large parts of Eastern Siberia. The analysis reveals a satisfactory agreement between modeled and measured components of the surface energy balance below the predominantly deciduous forest canopy. The mixed forest cover at our primary study site only contains 7% of deciduous larch trees. The LAI reduction implemented is therefore very small and had no noticeable effect on our results. This modification can be used to study further taxa-specific interactions with permanently frozen ground. It would also be desirable to implement a spatially-explicit, dynamic**

*vegetation model, such as the larch forest simulator (LAVESI, Kruse et al. (2016)) to further analyze the dynamic vegetation distribution under the recognition of the found interactions. This would allow us to simulate the vegetation response to changes in permafrost temperature and hydrology dynamically over a large timescale, and across a wide range of boreal forest ecosystems in Eastern Siberia."*

Page 8, line 8: How was soil thermal conductivity parameterised?

*We agree that this information should be added to our manuscript and have done so on p.9, l.195: "**Soil thermal conductivity is parameterised following Westermann et al. (2013 and 2016) and is based on the parameterization in Cosenza et al. (2003). The thermal conductivity of the soil is calculated as weighted power mean from the conductivities and volumetric fractions of the soil constituents water, ice, air, mineral and organic (Cosenza et al., 2003)."***

*Following Westermann et al. (2013, 2016) the according equation describing the soil thermal conductivity (k)*

$$k = \left( \sum_\alpha \Theta_\alpha \sqrt{k_\alpha} \right)^2$$

*is with the volumetric fractions (θα) of water, ice, air, mineral and organic. This parameterization has been used as standard in CryoGrid for a number of publications (i.e. Nitzbon et al. 2019, 2020). The temperature-dependence of the thermal conductivity, which gives rise to the thermal offset between ground surface and permafrost temperatures (Osterkamp and Romanovsky, 1999), is contained in the temperature-dependent water and ice contents as detailed in Sect. 2.2 in Westermann et al. (2013). This parameterization above is chosen for simplicity (e.g., de Vries, 1952; Farouki, 1981 describe other parameterizations), as reliable recommendations for a particular conductivity model are lacking for permafrost areas and as our chosen parameterization allows us to successfully reproduce observed annual freeze and thaw cycles at permafrost sites under differing environmental conditions (Westermann et al., 2013).*

Figure 3: What are the error bars showing? What is the input to calculate error bars (hourly, daily, weekly data)? It seems as if the variability (i.e. error bars) for the observed grassland turbulent energy fluxes is much larger than the modelled variability.

*We agree with the reviewer that this has not been made clear. Whiskers in modeled data show the standard deviation based on daily averaged*

*data. Further, the whiskers in the measured data at the grassland site were based on half-hourly data. We thank the reviewer for pointing this out and* **have adapted this for the measured data at the grassland site to be based on daily values as well.** *This provides more comparable variability values. We modify the figure caption accordingly to explain what the whiskers show (p.14):* **"Surface energy balance for snow-covered (28.10.2018-27.04.2019) and snow-free (10.10.2019-27.10.2019 and 28.04.2019- 10.10.2019) periods at the ground surface of grassland and forest and at the top of the canopy of forest (Forest TOC). Shown are the net radiation ($Q_{net}$), sensible ($Q_h$), latent ($Q_e$) and storage heat flux ($Q_s$) for the model runs of the forest and grassland site as well as the measured values at the grassland site. The bars indicate mean values while the whiskers show the corresponding standard deviations."**

Table 2: Is this the same data as shown in Figure 5? If so, the authors could think about showing only one of them (figure or table).

*We agree with the reviewer that this is indeed a repetition; therefore,* **the values shown in Table 2 are added to Figure 5 directly. Table 2 has been removed.**

Figure 6: Most of the model-observation comparison is of qualitative nature. The authors could add some performance metrics (e.g., RMSE, R2, bias: : :)

*We thank the reviewer for this suggestion and agree that performance metrics would be helpful to evaluate the model performance.* **We have added performance metrics (p-values) to describe the significance of the differences between the forest and grassland site to figure 5. We have adapted the caption accordingly (p.16): "Average measured and modeled snow-covered period GST, average measured and modeled snow-free period GST, annual average measured and modeled GST and the respective standard deviations in forest (top, at 0.03 m depth) and grassland (bottom, at 0.07 m depth)over a measurement period of 1 year (10. August 2018 - 10. August 2019). Unpaired t-test between modeled forest and grassland GST shows a statistically significant temperature difference for snow-covered and snow-free periods."**

Page 18, line 12: The authors report a bias in modelled GST during the winter. Since GST measurements are available, the forest-grassland comparison could be more meaningful if it were based on observational data.

*These numbers are provided in Figure 5 (and formerly Table 2). Based on this suggestion the measured differences will now also be provided*

*in the text (p. 18, l. 342):* **"Our results are in agreement with these observations, but further demonstrate that the impact of mixed boreal forest on the GST is strongest during the snow period and the summer peak with the warmest months. Our model reveals an average of 6.7°C higher GST during the snow-covered period and 5.2°C lower GST during the snow-free period. Measurements reveal an average of 2°C higher GST during the snow-covered period and 2.3°C lower GST during snow-free periods."**

Page 22, line 17: Could lateral flow of water contribute to differences in ground water content?

*We would like to thank the reviewer for this important question. Indeed lateral water flow could contribute to the differences in actual measured ground water content. This process is neglected in this baseline, one-dimensional model set-up where we try to investigate the influence of forest on the surface energy balance of the ground. Recently, Nitzbon et al. (2019) have integrated lateral fluxes of heat, water and snow in the CryoGrid scheme. Higher forested ground water content was measured in point measurements taken in 2018, but no year-long ground water content measurements are available for our forest site, therefore this is not discussed in detail within this study. We add this information to p.11, l.210:* **"Lateral water fluxes are neglected in this baseline, one-dimensional model set-up."** *We further add the following statement to the discussion (p. 22, l.441):* **"An aspect not represented in the model is the moisture transport and migration in frozen ground including the forming of ice lenses and excess ground ice, which can have a high impact on the local micro-topography and the surface energy balance. Furthermore, lateral water flow and snow redistribution may be important processes to be investigated in the future since they can strongly modify the ground thermal regime as well as the snow-pack development."**

In this manuscript, the authors developed a biophysical atmosphere-land interaction model (CryoGrid) and examined its performance by comparing with observations at a mixed forest in East Siberia. By coupling with a multi-layer canopy scheme (CLM-ml v0), they accounted for the effects of vegetation canopy on surface processes. They conducted a series of simulations with different land covers (forest and grassland) and different leaf area index or canopy density. They found that coverage with high leaf area index affects surface energy budget, such as solar radiation transfer and insulation, and that it could play important roles in snow and permafrost dynamics such as active layer thickness. I agree that terrestrial ecosystems in northern high latitude is remarkable in terms of climate change, especially permafrost thawing that is thought as one of the tipping elements of the Earth system. Then, developing advanced models simulating boreal ecosystem processes is highly important.

*We thank the reviewer for the positive perspective on the general topic of our study. We absolutely share the opinion that it is extremely relevant to study boreal forest and permafrost interactions under amplified climate change. Further, we gratefully acknowledge all critical comments which help us to improve our manuscript. We have thoroughly gone through all comments and suggestions made by the reviewer. Please note that any changes and additions we propose for the revised manuscript are highlighted in* **bold.**

Nevertheless, I could not find out what is the original contribution of this study. For example, the model developed in this study seems similar to previous models such as CLM, ORHIDEE, LPJ etc. These previous models have already implemented leaf phenology and dynamic vegetation; at least some versions include permafrost dynamics.

*We thank the reviewer for this critical remark on our manuscript and take this as an opportunity to clarify the scientific questions that we aim to answer with the performed modelling exercise. We appreciate your concern that previous models have already implemented leaf phenology or dynamic vegetation and would like to elaborate on this accordingly. The mentioned models such as Orchidee-Can (Chen et al. 2016), Lund-Potsdam-Jena (LPJ DGVM) (Beer et al. 2007), CLM (Levis et*

*al., 2004) or NEST (Zhang et al. 2003) and SiBCliM (Tchebakova et al. 2009) are discussed in the Introduction section on page 2 starting on line 22. Certainly these models include some sort of dynamic or static vegetation parameterization, leaf phenology and some models also permafrost dynamics. The focus thereof lies on the forest establishment and mortality (Sato et al. 2016), unfrozen vs. frozen ground and fire disturbances (Zhang et al. 2011) or the evolution of the vegetation carbon density under diverse warming scenarios (Beer et al. 2007). The multilayer canopy module presented here has been developed for a future integration in the CLM scheme, but has not been used in permafrost underlain boreal forests. Further, the presented model is an advancement as it is specifically developed to study heat transfer processes with both, high-resolution permafrost ground and a high-resolution vegetation canopy. This allows us to quantify the interactions between detailed canopy structures and permafrost. To amplify the importance and uniqueness of our study we add the following section to the introduction (p.2, l.50):* ***"While all of these studies have significantly improved our understanding of essential mechanisms in boreal permafrost ecosystems, it is important to further understand how a forest canopy affects the thermal state and the snow regime of the ground, especially amid ongoing shifts in forest composition (Loranty et al., 2018). The existing model set-ups are often static or not able to capture important processes such as the vertical canopy structure or the leaf physiological properties which determine the energy transfer between the top of the canopy atmosphere and the ground. To our knowledge, so far, none of the existing models is able to capture the important processes of the vertical canopy structure in combination with a physically-based, highly advanced permafrost model. The novel, physically-based model introduces a robust radiative transfer scheme through the canopy for a detailed analysis of the vegetation's impact on the hydro-thermal regime of the permafrost ground below. This allows us to quantify the surface energy balance dynamics below a complex forest canopy and its direct impact on the hydro-thermal regime of the permafrost ground below."***

*Further, we have made the conclusions and this study's original contribution clearer by adding the following key points (p.23, l.461):*

***"This study presents a specific application of a novel, coupled multilayer forest-permafrost model which enables us to investigate the energy transfer and surface energy balance in permafrost underlain boreal forest of Eastern Siberia. By simulating interactions between the vegetation cover and permafrost, our modeling approach allows us***

*to quantify and study the impact of the forest on the hydro-thermal regime of the permafrost ground below. An extensive comparison between measured and modeled energy balance variables (GST, $Q_e$, $Q_h$, $Q_{net}$, $S_{in}$ and $S_{out}$) reveals a satisfactory model performance justifying its application to investigate the thermal regime and surface energy balance in this complex ecosystem. Despite overall good performance, the field measurements reveal model shortcomings during the snow melt period. Based on this modeling exercise and field measurements, we investigate the thermal conditions of two landscape entities as they typically occur in the boreal zone. In regard to the forests insulation effect on permafrost and ongoing land cover transitions this study delivers important insights into the range of spatial differences and possible temporal changes that can be expected following landscape changes such as deforestation through fires, anthropogenic influences and afforestation in currently unforested grasslands or the densification of forested areas. The detailed vegetation model successfully calculates the canopy radiation and water budgets, leaf fluxes, as well as canopy turbulence and aerodynamic conductance. These canopy fluxes alter the below-canopy surface energy balance, the ground thermal conditions and the snow cover dynamics. We find a strong dampening effect of over 30°C on the annual ground surface temperature amplitude of the permafrost."*

Another concern on this study is that the authors used a very limited amount of observations. Especially, they validated the model performance to capture temporal variability only for surface temperature. Because many flux measurement sites are operating and providing a variety of observational data, I highly recommend validating the model by using a larger number of biophysical variables including energy and water fluxes.

*We appreciate the suggestion on using a larger number of biophysical variables including energy and water fluxes. In accordance with the comments of reviewer 1 we agree that our study requires a more extensive validation based on surface energy balance measurements. Thus we extended the model validation to an additional site for which extensive surface energy balance measurements are available in order to demonstrate the capabilities of our model. Not many such flux measurement sites are available in Eastern Siberia, to our knowledge no such site exists around our study side (Nyurba/Viluy area). We therefore use existing and available data from the rather near (581 km east), well-documented and well-studied research site Spasskaya-Pad at 62°14'N, 129°37'E. We have been provided meteorological and radiation data from beneath and above the larch-dominated forest canopy for 2018 through the Arctic Data Archive system (ADS). This data is used for*

*additional model validation. To justify our model we suggest adding this additional validation site to the appendix of our manuscript.*

*As described in our answer to reviewer 1 we have set-up and ran a 6-year simulation for this site, using ERA-interim forcing data for the coordinate above and a summer LAI of 1.9 and a tree height of 18 m. Since the study site is larch-dominated we have now implemented a simple leaf-off parameterisation which is used here. This results in a winter LAI of 0.5 and a leaf-off period from 10. October – 10. April.*

*We have added this additional model validation to the appendix of our manuscript since it is a rather technical aspect which is not directly related to our major study site. A detailed analysis of this validation run is presented in the following and in the revised manuscript. Detailed analyses show a good fit with the novel measurement data and therefore justify the use of the model in the current version. **We add the following paragraph to the section "Model validation and in-situ measurements" (p.13, l.270): "We acknowledge that the target variable GST does not provide a full picture of the surface energy balance. Therefore, we further validate the model performance with additional measurements from an external study site. Through the Arctic Data Archive system (ADS) we have been provided meteorological and radiation data from beneath and above the larch-dominated forest canopy in Spasskaya-Pad for 2017-2018 (Maximov et al., 2019). This data is used for additional model validation and is added to the appendix of our manuscript. Overall our analysis reveals a satisfactory agreement between modeled and measured components of the surface energy balance below the canopy. Thus, we argue that the performance of the model at the external study site justifies its application at the primary study site in Nyurba where below canopy fluxes were not acquired (see Appendix C)."***

*Respectively, we add the following paragraph to Appendix C (p.27):*
***Further validation of the model performance is performed for a well-studied research site in Spasskaya-Pad at N 62.14°, E 129.37°. This additional validation site is located 581 km from our primary study site, but allows to validate further model variables due to additional observational data. Spasskaya-Pad is a continuous permafrost region and the active layer depth is about 1.2 m in larch-dominated forests. Main tree species is Dahurian larch (Larix gmelinii) with a stand density of 840 trees/ha. Understory vegetation (Vaccinium) is dense and 0.05 m high. In 1996 a 32m observation tower was installed (Ohta et al., 2001) in larch dominated forest. Through***

*the Arctic Data Archive system (ADS, https://ads.nipr.ac.jp/) we have been provided with the most recent, available meteorological and radiation data from beneath and above the larch-dominated forest canopy for the time-period 2017-2018 (Maximov et al., 2019). Each variable used here is measured at exactly 5-min intervals, except radiation. Ventilated shelters cover air temperature and humidity sensors. Net all-wave radiation and four components of radiation are measured every minute, and the data loggers record average,maximum and minimum values. Upward and downward long-wave radiation is corrected using the sensed temperature at domes and sensor bodies. Ground temperature is measured at seven depths, and soil moisture at five depths. A more detailed description of the sensors can be found in Table 1 in Ohta et al. (2001). We have set-up and ran a 6-year simulation for this study site (2013-2019), using ERA-interim forcing data using the grid cell closest to the coordinate N 62.14°, E 129.37°. The measurement tower is situated in larch-dominated forest and we have now implemented a simple leaf-off parameterisation which is used here. Based on Ohta et al. (2001) we define a partial leaf-off period from 10. October - 30. May, resulting in a winter LAI of 0.5 $m^2m^{-2}$. Summer LAI is 1.9 $m^2m^{-2}$ and a tree height of 18 m. For a comprehensive model validation we use the same soil parameters for this external study site. Table A3 summarizes all parameter choices for soil stratigraphies and Table A4 summarizes constants used. The subsurface stratigraphy extends to 100 m below the surface where the geothermal heat flux is set to 0.05 $W/m^{-2}$ (Langer et al., 2011b). The ground is divided into separate layers in the model. The uppermost 8 m have a layer thickness of 0.05 m, followed by 0.1 m for the next 20 m, 0.5 m up to 50 m and 1 m there after. The remaining CryoGrid parameters were adopted from previous studies using CryoGrid (Table A2) (Langer et al., 2011a, b, 2016; Westermann et al., 2016; Nitzbon et al., 2019, 2020). Again, we use ground surface temperature (GST) as one of the target variables for model validation, measured at 0.2 m. In addition we use air temperature below the canopy, measured at the height of 1.2 m, net radiation ($Q_{ne}t$), latent ($Q_e$) and sensible ($Q_h$) heat flux, and incoming ($S_{in}$) and outgoing ($S_{out}$) shortwave radiation flux at the ground surface as target variables allowing a comprehensive validation of the modeled heat and moisture exchange processes within and below the canopy.*

*We assess the surface energy balance by comparing the median weekly values of modeled and measured net radiation (Qnet), sensible heat flux ($Q_h$), latent heat flux ($Q_e$), and incoming ($SW_{in}$) and outgoing ($SW_{out}$) solar radiation at the forested site (Fig. C1).*

*Modeled turbulent fluxes at forest ground are small for snow-covered periods and measured data are not available during this period. Modeled and measured sensible heat flux in the snow-free period differ by 0.1 $Wm^{-2}$ only. Modeled latent heat flux is only a fourth of the measured value and therefore underestimated in our model. Modeled net radiation in the snow-free period (25.7 $Wm^{-2}$) is slightly above measured net radiation (19.4 $Wm^{-2}$). For the snow-covered period modeled net radiation is slightly below the measured median value. The incoming shortwave radiation measured and modeled for the forest site fit well with differences well below 10 $Wm^{-2}$. The standard deviation of measured values is higher for all variables except snow-covered net radiation.*

*In a second step, we compare the modeled and measured annual, snow-free and snow-covered median GST and air temperature below the canopy to understand the overall model performance regarding the thermal regime of the surface and the ground and the relative temperature differences between the model and measurements (Fig. C2). The highest deviation between modeled and measured temperatures is found in the GST of the snow-free period. Here, the model shows a cold bias of 2°C. For the snow-covered period the difference is 1.8°C. For the air temperature below the canopy the difference between modeled and measured in the snow-free period is 1.5°C, for snow-covered the difference is again 1.8°C. This falls into the range of 1.5-2°C that are commonly used for validation purposes (Langer et al., 2013, Westermann et al., 2016).*

*Overall our analysis reveals a satisfactory agreement between modeled and measured components of the surface energy balance below the canopy. Thus, we argue that the performance of the model at the external study site justifies its application at the primary study site in Nyurba where below canopy fluxes were not acquired."*

*Fig. C1 and Fig. C2 are added with the following caption for Fig. C1: "Modeled (grey) incoming and outgoing solar radiation ($S_{in}$, $S_{out}$) and turbulent fluxes ($Q_h$, $Q_e$, $Q_{net}$) for snow-covered (28.10.2017-27.04.2018, above) and snow-free (10.10.2017-27.10.2017 and 28.04.2018-10.10.2018, below) periods at the ground surface of forest." and for Fig. C2: "Left, modeled (grey) and measured (black) air temperature (°C) below the canopy. Right, ground surface temperature (°C). Both for snow-covered (28.10.2017-27.04.2018, above) and snow-free (10.10.2017-27.10.2017 and 28.04.2018-10.10.2018, below) periods at the forest site in Spasskaya Pad."*

Also, I have a concern about the overly simplified simulation conditions, such as the lack of seasonal change in leaf area index in forest. Though in situ data were not available, at present, we can obtain time-series of leaf area index from satellite remote sensing as a good proxy.

*We thank the reviewer for this critical remark concerning the canopy phenology. We discuss the implementation of the needle-tossing later in the manuscript (p.23, l.449):* **"To simulate the needle-tossing of deciduous larch, we have incorporated a leaf area index threshold between needle-tossing and leaf-out (10. October - 30. May) for simulations at our external validation site at Spasskaya-Pad (see Appendix C). This tunes the model towards a more detailed representation of larch-dominated forests, which are particular to the secondary study site and large parts of Eastern Siberia. The analysis reveals a satisfactory agreement between modeled and measured components of the surface energy balance below the predominantly deciduous forest canopy. The mixed forest cover at our primary study site only contains 7% of deciduous larch trees. The LAI reduction implemented is therefore very small and had no noticeable effect on our results. This modification can be used to study further taxa-specific interactions with permanently frozen ground. It would also be desirable to implement a spatially-explicit, dynamic vegetation model, such as the larch forest simulator (LAVESI, Kruse et al. (2016)) to further analyze the dynamic vegetation distribution under the recognition of the found interactions. This would allow us to simulate the vegetation response to changes in permafrost temperature and hydrology dynamically over a large timescale, and across a wide range of boreal forest ecosystems in Eastern Siberia."**

Based on the inadequacy of scientific insights and model simulations, I cannot recommend this manuscript to accept for publication in the present form. The manuscript is too descriptive and needs more model validation with observations. Also, the authors need to devise model simulations to make insightful discussions.

We thank the reviewer for her/his opinion on our submitted manuscript and would like to summarize the substantial changes and adaptations described in detail above and in the responses to Reviewer 1. The main changes we propose to improve our manuscript are:

- Adding a further validation site (Spasskaya Pad) with an extensive record of energy flux measurements which is, to our knowledge, a unique data record for a forested study site in Eastern Siberia. Due to a higher component of deciduous taxa we

have implemented a canopy phenology module to simulate the specific traits of deciduous taxa. With this additional validation site we justify the use of our current model set-up.
- We would also like to point out changes in the introduction and conclusion sections (see above) which now clarify the novelty of our modeling exercise and the importance of the found and described insights in severely understudied high-latitude, permafrost-underlain boreal forest areas.

Minor points
Page 3 Section 2.1: Can you give information on vegetation conditions, such as leaf area index and tree density?

*Following our response to reviewer 1, we recognize that the information given on LAI estimation on p.12, l.230 is insufficient, therefore we have modified the paragraph to the following, more detailed description:* **"LAI can be estimated from satellite data, calculated from below-canopy light measurements or by harvesting leaves and relating their mass to the the canopy diameter. Ohta et al. (2001) have described the monitored deciduous-needleleaf forest site at Spasskaya Pad research station, which has comparable climate conditions but is larch-dominated. The value of the tree plant area index (PAI), obtained from fish-eye imagery and confirmed by litter fall observations, varied between 3.71 in the foliated season and 1.71 in the leafless season. This value does not include the ground vegetation cover. Further, Chen et al. (2005) compared ground-based LAI measurements to MODIS values at an evergreen-dominated study area (57.3° N, 91.6° E) south-west of the region discussed here, around the city of Krasnoyarsk. The mixed forest consists of spruce, fir, pine and some occasional hardwood species (birch and aspen). They find LAI values between 2 and 7. To assess the LAI we use data from literature and the experience from the repeated field work at the described site. Following Kobayashi et al. (2010) who conducted an extensive study using satellite data, the average LAI for our forest type is set to 4 $m^2m^{-2}$ and stem area index (SAI) is set to 0.05 $m^2m^{-2}$, resulting in a plant area index (PAI)of 4.05 $m^2m^{-2}$ and 9 vegetation layers for model simulations."** *To further answer this question, leaf area index has not been measured explicitly, but is described in the study by Kobayashi et al. (2010) and can also be estimated from satellite imagery such as the Copernicus LAI 300m Version 1 product (available through https://land.copernicus.vgt.vito.be/PDF/portal/Application.html#Home).*

*Accordingly, LAI at our study site is between 3 and 4 $m^2/m^2$. Tree density could be estimated from the vegetation survey mentioned on page 5, line 20.*

Page 7 Line 3: I am not sure how within-canopy wind profile was parameterized and simulated by the multi-layer canopy model. Can you explain briefly, because this is an important feature of multi-layer canopy models?

*This is not discussed in particular here, but described in detail in Bonan et al. 2018. We add this information to our manuscript (p.7,l.154): "**The within-canopy wind profile is calculated using above- and within-canopy coupling with a roughness sublayer (RSL) parameterization (see Bonan et al. 2018 for further detail).**"*

**Marked-up manuscript version**

[revised manuscript text omitted]

---

## Referee Report (RR1)

The study of Stuenzi et al. contribute to important discussion how the vegetation cover influences the surface energy balance and thus thawing of permafrost and a possible feedback to the atmosphere. The authors have shown how the different vegetation cover influences the ground surface temperature and top canopy surface energy balance , but they have waived to show the difference in active layer thickness which could increase the importance of the study.

The authors have answered very detailed to the reviews, which I really appreciate. They have included one additional validation site. Even so, I would recommend further efforts to validate the simulations against measurements. For the forest stands only a short period of ground surface temperature is evaluated. It is interesting to see that the assumptions lead to large differences of the ground and top canopy energy budget, which unfortunately is not underpinned with measurements. Looking at the surface energy balance, I doubt the significance of the difference between forest TOC and grassland, as differences between modeled and measured grassland are similar.

In general, I would also appreciate further details on the various assumptions made. Most of the methods used are described by using reference but the reader need to go through all the different studies to which the authors refer to. For example, how is the heat flux within the soil layer calculated, as this is of great importance in permafrost regions. Nonetheless, I would like to point out that this study is very well structured and written, which enables the reader to follow the manuscript very easily.

---

## Author Response (AR2)

Reviewer comments

*Author response,* ***Changes made in manuscript***

**-----------------------**

**Reviewer comment on "Variability of the Surface**

**Energy Balance in Permafrost Underlain Boreal**

**Forest" by Simone Maria Stuenzi et al.**

**Anonymous Referee #3**

Received: 16 November 2020

The study of Stuenzi et al. contribute to important discussion how the vegetation cover influences the surface energy balance and thus thawing of permafrost and a possible feedback to the atmosphere.

*We thank the reviewer for the positive general evaluation of our study and for taking the time to review our manuscript. We have worked through all of the posed questions and suggestions made, which has further improved our manuscript. Please note that any changes and additions to the text that we propose are highlighted in* ***bold.***

*With this study, we indeed try to advance our understanding of the vegetation cover's impact on the surface energy balance, the thermal development of permafrost and the related atmospheric feedback. We would like to thank the reviewer for acknowledging the significance of this modeling study in helping us further understand permafrost responses to a changing climate.*

The authors have shown how the different vegetation cover influences the ground surface temperature and top canopy surface energy balance, but they have waived to show the difference in active layer thickness which could increase the importance of the study.

*We agree with the reviewer that the active layer thickness is one of the most important variables in permafrost-ecosystems. Nevertheless, we think that this has not been made clear enough in our manuscript. Therefore we add the following statement in order to stress the importance of the ALT on p.15,l.305: "In a second step, we compare the annual, snow-free and snow-covered period average GST to understand the overall model performance and the relative temperature differences between the forest and grassland sites (see Fig. 5).* ***We further***

*discuss the annual cycle of the thermal development of the permafrost ground, the modeled and measured active layer thickness, as well as the volumetric ground water content at the forest and grassland sites.*"

*Active layer thickness measurements and the modeled values for the grassland and forest sites are discussed on p.17,l.317: "To further investigate the temporal evolution of the permafrost ground, we compare modeled and measured active layer thicknesses at both study sites. In the grassland, the modeled maximum active layer thickness (ALT) is 2.35 m between 13. and 24. October 2018, complete freezing occurred on the 9th of November and top soil thawing started on the 3rd of May. The measured ALT in the grassland was 2.3 m in mid-August 2018 and early-August 2019. The measured ALT in the forest was between 0.5 and 1.1 m in mid-August 2018. In the forest, the modeled maximum ALT is 2.05 m in October 2018 with freezing being completed on the 14th of November. Top soil thawing begins on June 23rd, 51 days later than in grassland. The modeled ALT in August 2018 is between 0.4 m and 1.8 m and therefore overestimated by 0.3 m compared to the point measurements taken in August 2018 (see Fig. 7)."*

**In addition, we have now added the modeled ALT dynamics and the point measurements taken in 2018 and 2019 and described above to Figure 7. We have updated the figure caption accordingly: "Left, Measured snow depth at AWS (black solid line) in the grassland, modeled snow depth in grassland (dashed light green) and modeled snow depth in forest (dashed dark green). Right, Active layer thickness (ALT) dynamics, measured ALT at the grassland and forest sites (black, point measurements in 2018 and 2019 (grassland only)), modeled ALT in forest (dashed light green) and modeled ALT in grassland (dashed dark green) sites. Top soil freezing starts at the beginning of October."**

The authors have answered very detailed to the reviews, which I really appreciate. They have included one additional validation site. Even so, I would recommend further efforts to validate the simulations against measurements. For the forest stands only a short period of ground surface temperature is evaluated.

*We thank the reviewers and the editor for all previous comments and suggestions, which greatly improved our manuscript. We think that the additional validation site significantly improves the model validation and demonstrates the abilities of the coupled permafrost-multilayer vegetation model for application in permafrost underlain boreal forest ecosystems.*

*We further thank the reviewer for the critical remark on additional efforts to validate the model. We would like to take this as an opportunity to clarify the validity of the model results further and appreciate the suggestion on using further measurements. Due to the rarity of measurement datasets in this area and the challenges of setting up long-term measurements, we have decided to use the closest, available dataset for the additional model validation. At the Spasskaya-Pad research station, we have been able to show the validity*

*of our model results by comparing the simulation results to one whole year of measurements. We compare the ground temperature and air temperature below canopy, as well as the net radiation, incoming and outgoing shortwave radiation, and latent and sensible heat fluxes at the ground surface below the forest canopy.*

*At our primary study site, we have below-canopy ground surface temperature measurements for one entire year (August 2018 - August 2019). This dataset, in combination with the AWS at the grassland site, is used for model validation at the two forest sites and grassland sites. In the scope of this study, we focus on demonstrating the annual dynamics of the surface energy balance. We are convinced that in this context the performed model validation is sufficient. Additionally, we would like to point out that considering the entire annual cycle is great advancement over previous studies which focus on either the summer or the winter surface energy balance, only (i.e. the study by Otha et al. (2001) which was conducted at our secondary study site Spasskaya-Pad). Furthermore, we would like to point out that the model runs with a spin-up period of five years before being compared to the field measurements of the year 2018/2019. The model validation is thus performed under dynamic conditions.*

*We agree that for a study with a focus on long-term predictions or on the impact of climate warming longer validation time-series would be necessary. Field measurements are the key component to getting a better handle on the diverse processes involved in the functioning of this complex ecosystem. Therefore, further data acquisition and integration of additional datasets is discussed as part of the model limitations on p.22,l.437: "This points to more complex processes that control snow melt in forest than currently represented by the model. Thus, it would be highly desirable to obtain further field measurements in order to gain a better understanding of snow melt processes in boreal forests.", and on p.23,l.445: "Additionally, more detailed field studies and modeling exercises on the variation of canopy densities and structures should be carried out in order to obtain a better understanding of the impact of dynamic forest stand development on permafrost and vice versa."*

It is interesting to see that the assumptions lead to large differences of the ground and top canopy energy budget, which unfortunately is not underpinned with measurements. Looking at the surface energy balance, I doubt the significance of the difference between forest TOC and grassland, as differences between modeled and measured grassland are similar.

*The reviewer correctly notices that we do not have above canopy measurements to compare the TOC energy budget to. We would like to acknowledge that the focus of this study is on heat exchange processes between the forest and the ground. We argue that the model is best validated exactly at this interface. We, thus, thoroughly validate the model with ground surface temperature measurements at all sites and below canopy radiation measurements at our secondary study site. We further argue that below canopy radiation measurements implicitly include radiative absorption and reflection within the canopy so that*

*the below canopy net radiation provides a good measure whether the TOC energy balance is well represented by our model.*

*Regarding the significance of energy flux differences we clearly demonstrate that the differences between forest TOC and forest ground are much higher (around 50 Wm$_{-2}$) than the differences between forest TOC and grassland (around 10 Wm$_{-2}$). We further demonstrate that our model is able to reproduce the below canopy SEB and the SEB at the grassland site. This gives us confidence that our model also realistically reproduces the differences in storage heat flux and net radiation flux between forest TOC and forest ground. Further evidence that the model realistically reproduces the heat flux partition is given by the good agreement between measured and modeled ground surface temperatures and the active layer thicknesses.*

*All model components are validated thoroughly based on multiple variables over the entire annual cycle. We are convinced that this allows us to evaluate the modeled systematic differences in the surface energy balances between the forested and the grassland site.*

In general, I would also appreciate further details on the various assumptions made. Most of the methods used are described by using reference but the reader need to go through all the different studies to which the authors refer to. For example, how is the heat flux within the soil layer calculated, as this is of great importance in permafrost regions.

*We thank the reviewer for pointing out these unclarities. To avoid further lengthening of this manuscript and keeping it readable, we have decided against providing further model details, which are not directly changed or modified in this study, but to kindly refer to the previous CryoGrid publications. CryoGrid has been used and further developed in many different studies and i.e. the heat flux between below ground soil layers is explained in detail in the well-documented CryoGrid model publication by Westermann et al. (2016), published in the model development journal GMD.*

*Further, we have previously added the following paragraph to our methods section, following the recommendations of reviewer 1 and 2. The soil layer conductivity is now explained on p.9,l.194: "Soil thermal conductivity is parameterized following Westermann et al.(2013, 2016) and is based on the parameterization in Cosenza et al. (2003). The thermal conductivity of the soil is calculated as weighted power mean from the conductivities and volumetric fractions of the soil constituents water, ice, air, mineral and organic (Cosenza et al., 2003)."*

Nonetheless, I would like to point out that this study is very well structured and written, which enables the reader to follow the manuscript very easily.

*We would like to thank the reviewer for this closing statement and the help on improving our manuscript. We would again also like to thank all reviewers and the editor for guiding us towards this well-structured and easy-to-follow manuscript.*

**List of all relevant changes made in the manuscript:**

1.  We have added the active layer thickness dynamics to Figure 7 and have updated the caption accordingly.

**Marked-up manuscript version**

[revised manuscript text omitted]